# Interpersonal alignment of neural evidence accumulation to social exchange of confidence

Jamal Esmaily[1,2,3]\*, Sajjad Zabbah[4,5,6], Reza Ebrahimpour[7]\*[†], Bahador Bahrami[1,8]\*[†]

[1]Department of General Psychology and Education, Ludwig Maximillian University, Munich, Germany; [2]Faculty of Computer Engineering, Shahid Rajaee Teacher Training University, Tehran, Islamic Republic of Iran; [3]Graduate School of Systemic Neurosciences, Ludwig Maximilian University Munich, Munich, Germany; [4]School of Cognitive Sciences, Institute for Research in Fundamental Sciences (IPM), Tehran, Islamic Republic of Iran; [5]Wellcome Centre for Human Neuroimaging, University College London, London, United Kingdom; [6]Max Planck UCL Centre for Computational Psychiatry and Aging Research, University College London, London, United Kingdom; [7]Institute for Convergent Science and Technology, Sharif University of Technology, Tehran, Islamic Republic of Iran; [8]Centre for Adaptive Rationality, Max Planck Institute for Human Development, Berlin, Germany

**\*For correspondence:**
jimi.esmaily@gmail.com (JE);
ebrahimpour@sharif.edu (RE);
bbahrami@gmail.com (BB)

[†]These authors contributed equally to this work

**Competing interest:** The authors declare that no competing interests exist.

**Abstract** Private, subjective beliefs about uncertainty have been found to have idiosyncratic computational and neural substrates yet, humans share such beliefs seamlessly and cooperate successfully. Bringing together decision making under uncertainty and interpersonal alignment in communication, in a discovery plus pre-registered replication design, we examined the neuro-computational basis of the relationship between privately held and socially shared uncertainty. Examining confidence-speed-accuracy trade-off in uncertainty-ridden perceptual decisions under social vs isolated context, we found that shared (i.e. reported confidence) and subjective (inferred from pupillometry) uncertainty dynamically followed social information. An attractor neural network model incorporating social information as top-down additive input captured the observed behavior and demonstrated the emergence of social alignment in virtual dyadic simulations. Electroencephalography showed that social exchange of confidence modulated the neural signature of perceptual evidence accumulation in the central parietal cortex. Our findings offer a neural population model for interpersonal alignment of shared beliefs.

## Editor's evaluation

This important study examines how humans use information about the confidence of collaborators to guide their own perceptual decision making and confidence judgements. The study addresses this question with a combination of psychophysics, electrophysiological modeling, and computational modelling that provides a compelling validation of a computational framework that can be used to derive and test theory-based predictions about how collaborators use communication to align their confidence and thereby optimize their collective performance.

## Introduction

We communicate our confidence to others to share our beliefs about uncertainty with them. However, numerous studies have shown that even the same verbal or numerical expression of confidence can

have very different meanings for different people in terms of the underlying uncertainty (*Ais et al., 2016*; *Navajas et al., 2017*; *Fleming et al., 2010*). Similar inter-individual diversity has been found at the neural level (*Fleming et al., 2010*; *Sinanaj et al., 2015*; *Baird et al., 2013*). Still, people manage to cooperate successfully in decision making under uncertainty (*Bahrami et al., 2010*; *Austen-Smith and Banks, 1996*). What computational and neuronal mechanisms enable people to converge to a *shared meaning* of their confidence expressions in interactive decision making despite the extensively documented neural and cognitive diversity? This question drives at the heart of recent efforts to understand the neurobiology of how people adapt their communication to their beliefs about their interaction partner (*Stolk et al., 2016*). A number of studies have provided compelling empirical evidence of brain-to-brain coupling that could underlie adaptive communication of shared beliefs (*Silbert et al., 2014*; *Honey et al., 2012*; *Hasson et al., 2004*; *Dikker et al., 2014*; *Konvalinka et al., 2010*). These works remain, to date, mostly observational in nature. Plausible neuro-computational mechanism(s) accounting for how interpersonal alignment of beliefs may arise from the firing patterns of decision-related neural populations in the human brain are still lacking (*Hasson and Frith, 2016*; *Wheatley et al., 2019*). Using a multidisciplinary approach, we addressed this question at behavioral, computational, and neurobiological levels.

By sharing their confidence with others, joint decision makers can surpass their respective individual performance by reducing uncertainty through interaction (*Bahrami et al., 2010*; *Sorkin et al., 2001*). Recent works showed that during dyadic decision making, interacting partners adjust to one another by matching their own average confidence to that of their partner (*Bang et al., 2017*). Such confidence matching turns out to be a good strategy for maximizing joint accuracy under a range of naturalistic conditions, e.g., uncertainty about the partner's reliability. However, at present there is no link connecting these socially observed emergent characteristics of confidence sharing with the elaborate frameworks that shape our understanding of confidence in decision making under uncertainty (*Navajas et al., 2017*; *Fleming et al., 2010*; *Pouget et al., 2016*; *Adler and Ma, 2018*; *Aitchison et al., 2015*).

Theoretical work has shown that sequential sampling can, in principle, provide an optimal strategy for making the best of whatever uncertain, noisy evidence is available to the agent (*Heath, 1984*). These models have had great success in explaining the relationship between decision reaction time (RT) and accuracy under a variety of conditions ranging from perceptual (*Hanks and Summerfield, 2017*; *Gold and Shadlen, 2007*) to value-based decisions (*Ruff and Fehr, 2014*) guiding the search for the neuronal mechanisms of evidence accumulation to boundary in rodent and primate brains (*Schall, 2019*). The relation between RT and accuracy, known as speed-accuracy trade-off, has been recently extended to a three-way relationship in which choice confidence is guided by *both* RT and probability (or frequency) of correct decision (*Pouget et al., 2016*; *Kiani et al., 2014*; *Vickers, 1970*). Critically, these studies have all focused on decision making in *isolated individuals* deciding privately (*Wheatley et al., 2019*). Little is known about how these computational principles and neuronal mechanisms can give rise to socially shared beliefs about uncertainty.

To bridge this gap, we examined confidence-speed-accuracy trade-off in social vs isolated context in humans. We combined a canonical paradigm (i.e. dynamic random dot motion [RDM]) extensively employed in psychophysical and neuroscientific studies of speed-accuracy-confidence trade-off (*Hanks and Summerfield, 2017*; *Gold and Shadlen, 2007*; *Kelly and O'Connell, 2013*) with interactive dyadic social decision making (*Bahrami et al., 2010*; *Bang et al., 2017*). We replicated the emergence of confidence matching and obtained pupillometry evidence for shared subjective beliefs in our social implementation of the random dot paradigm and we observed a novel pattern of confidence-speed-accuracy trade-off specifically under the social condition. We constructed a neural attractor model that captured this trade-off, reproduced confidence matching in virtual social simulations and made neural predictions about the coupling between neuronal evidence accumulation and social information exchange that were born out by the empirical data.

## Results

We used a discovery-and-replication design to investigate the computational and neurobiological substrates of confidence matching in two separate steps: 12 participants (4 female) were recruited in study 1 (discovery) and 15 (5 female, age: 28 (mean) ± Std (7)) in study 2 (replication, second study was pre-registered: https://osf.io/5zces). In each study, participants reported the direction of a random-dot

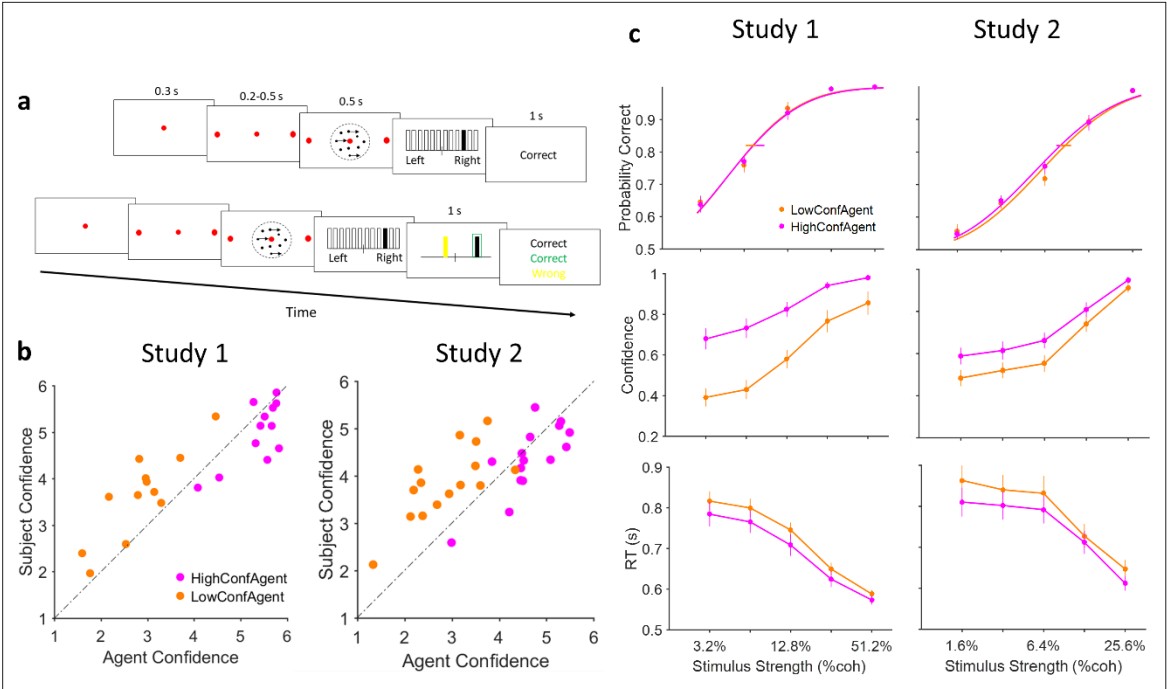

**Figure 1.** Experiment paradigm and behavioral results. (**a**) Timeline of trials in isolated (top) and social (bottom) conditions. After stimulus presentation, subjects reported their decision and confidence simultaneously by clicking on 1 of the 12 vertical bars. In the social condition, decision and confidence of participant (white in the experiment, here black for illustration purpose) and partner (yellow) were color coded. (**b**) Confidence matching. Participants confidence against agent confidence show a significant relation in both studies (linear regression p<0.001 for both studies). (**c**) Under social condition, when participants were paired with high (magenta) vs low (dark orange) confidence partner, accuracy (top panel) did not change (horizontal lines, 68% confidence interval of bootstrap test with 10,000 repetitions) but confidence (middle panel) and reaction time (RT) (bottom panel) were altered. Curves fitted to the accuracy data are Weibull cumulative distribution function. Error bars are standard error of the mean (SEM) across subjects.

The online version of this article includes the following figure supplement(s) for figure 1:

**Figure supplement 1.** Accuracy and confidence of the computer generated partners (CGPs).

**Figure supplement 2.** Statistical analysis of the confidence matching effect.

**Figure supplement 3.** Examination of the hypothesis that the partner's confidence at trial *t* modulates the participant behavior at trial *t+1*.

**Figure supplement 4.** Summary of debriefing results of the second study.

motion stimulus and indicated their confidence (*Figure 1a*) while EEG and eye tracking data were recorded, simultaneously. After an extensive training procedure (see Materials and methods for the recruitment), participants reached a stable behavioral (accuracy and RT) performance level. Then, two experimental sessions were conducted: first a private session (200 trials) in which participants performed the task alone; then a social session (800 trials for study 1 and 400 for study 2) in which they performed the task interactively together with a partner (implied to be another participant in a neighboring lab room).

In every trial (*Figure 1a*), after fixation for 300 ms was confirmed by closed-loop real-time eye tracking, two choice-target points appeared at 10° eccentricity corresponding to the two possible motion directions (left and right). After a short random delay (200–500 ms, truncated exponential distribution), a dynamic RDM (see *Shadlen and Newsome, 2001*) was centrally displayed for 500 ms in a virtual aperture (5° diameter). At the end of the motion sequence, the participant indicated the direction of motion and their confidence on a 6-point scale by a single mouse click. A horizontal line intersected at midpoint and marked by 12 rectangles (6 on each side) was displayed. Participants moved the mouse pointer – initially set at the midpoint – to indicate their decision (left vs right of midpoint) and confidence by clicking inside one of the rectangles. Further distance from the midpoint indicated more confidence. RT was calculated as the time between the onset of the motion stimulus sequence and the onset of deviation of the mouse pointer (see Materials and methods for more details) (*Resulaj et al., 2009*) at the end of stimulus presentation.

In the isolated trials, the participant was then given visual feedback for accuracy (correct or wrong). In the social trials (*Figure 1a*, bottom panel), after the response, participants proceeded to the social stage. Here, the participants' own choice and confidence as well as that of their partner were displayed coded by different colors (white for participants; yellow for partners). Joint decision was automatically arbitrated in favor of the decision with higher confidence. Finally, three distinct color-coded feedback messages (participant, partner, and joint decision) were displayed.

Participants were instructed to try to maximize the joint accuracy of their social decisions. In order to achieve joint benefit, confidence should be expressed such that the decision with higher probability of correct outcome dominates (*Bahrami et al., 2010*). For this to happen, the participant needs to factor in the partner's behavior and adjust her confidence accordingly. For example, if the participant believes that her decision is highly likely to be correct, her confidence should be expressed such that joint decision is dominated by the partner only if the probability that the partner's decision is correct is even higher (and not, for example, if the partner expressed a high confidence habitually). This social modulation of one's confidence in a perceptual decision comprises the core of our model of social communication of uncertainty.

Following from an earlier study (*Bang et al., 2017*), for each block the participants were led to believe that they were paired with a new, anonymous human partner. In reality, in separate blocks, they were paired with four computer generated partners (henceforward, CGPs; see Materials and methods) constructed and tuned to parameters obtained from the participant's own behavior in the isolated session: (1) high accuracy and high confidence (HAHC; i.e. this CGP's decisions were more likely to be more confident as well as more accurate); (2) high accuracy and low confidence (HALC); (3) low accuracy and high confidence (LAHC); and (4) low accuracy and low confidence (LALC) (see Materials and methods for details). For study 2, we used two CGPs (HCA and LCA) while the agent accuracy was similar to those of participants (*Bang and Fleming, 2018*) (Wilcoxon rank sum, p=0.37, $df$ = 29, $zval$ = 0.89). See *Figure 1—figure supplement 1* for confidence and accuracy data of CGPs. Each participant completed 4 blocks of 200 trials cooperating with a different CGP in each block. Our questionnaire results also confirmed that our manipulation indeed worked (*Figure 1—figure supplement 4*) and more importantly none of the subject suspected their partners was an artificial one.

Having observed the confidence matching effect in both studies (*Figure 1b*), a permutation analysis confirmed that this effect did not arise trivially from mere pairing with any random partner (*Bang et al., 2017*; *Figure 1—figure supplement 2*). The difference between the participant's confidence and that of their partner was smaller in the social (vs isolated) condition (*Figure 1—figure supplement 2*) consistent with the prediction that participants would match their average confidence to that of their partner in the social session (*Bang et al., 2017*).

Having established the socially emergent phenomenon of confidence matching in the dynamic RDM paradigm, we then proceeded to examine choice speed, accuracy, and confidence under social

**Table 1.** Details of statistical results in behavioral data (*Figure 1*).

|  | Response | Regressors | Estimate | SE | CI | t-Stat | p-Value | Total number |
|---|---|---|---|---|---|---|---|---|
| **Study 1** | Accuracy (HC vs LC) | Coherency | 0.007 | 0.0006 | [0.006 0.008] | 11.57 | <0.001 | 9600 |
|  |  | Condition | −0.002 | 0.021 | [−0.045 0.04] | −0.1 | 0.92 | 9600 |
|  | Confidence (HC vs LC) | Coherency | 0.0475 | 0.0008 | [0.046 0.049] | 56.5 | <0.001 | 9600 |
|  |  | Condition | 1.361 | 0.03 | [1.31 1.42] | 46.4 | <0.001 | 9600 |
|  | RT (HC vs LC) | Coherency | −0.005 | 0.0001 | [−0.005 −0.004] | −44.4 | <0.001 | 9600 |
|  |  | Condition | 0.029 | 0.004 | [−0.035 −0.021] | 7.85 | <0.001 | 9600 |
| **Study 2** | Accuracy (HC vs LC) | Coherency | 0.0209 | 0.0016 | [0.017 0.024] | 13.23 | <0.001 | 6000 |
|  |  | Condition | −0.0092 | 0.0296 | [−0.067 0.049] | −0.31 | 0.76 | 6000 |
|  | Confidence (HC vs LC) | Coherency | 0.1011 | 0.1011 | [0.097 0.106] | 47.47 | <0.001 | 6000 |
|  |  | Condition | 0.496 | 0.037 | [0.42 0.56] | 13.32 | <0.001 | 6000 |
|  | RT (HC vs LC) | Coherency | −0.009 | 0.0003 | [−0.01 −0.008] | −26.22 | <0.001 | 6000 |
|  |  | Condition | 0.0363 | 0.006 | [0.024 0.048] | 6.12 | <0.001 | 6000 |

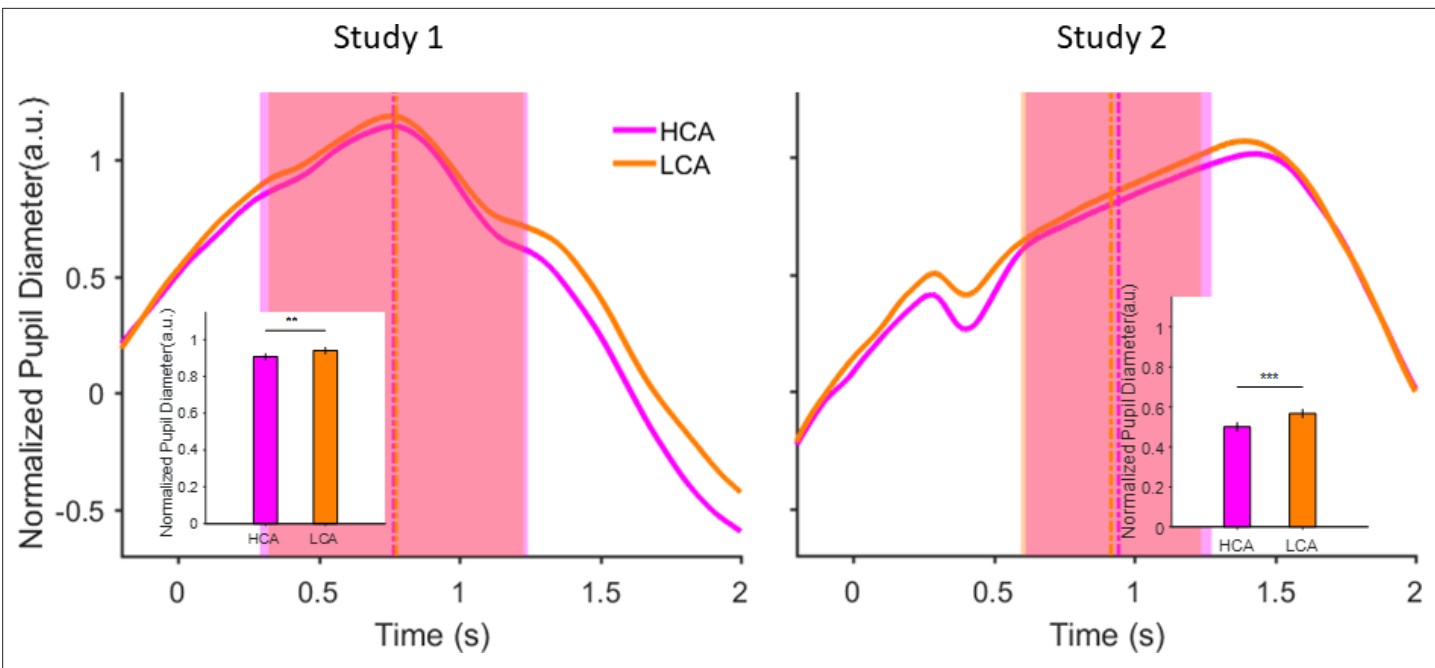

**Figure 2.** Pupil size during inter-trial interval (ITI) under pairing conditions in the social context when participant was paired with a high (HCA) or low confidence (LCA) agent. Normalized pupil diameter aligned to start of ITI period (*t*=0). Vertical dashed lines show average ITI duration. The shaded areas are one standard deviation of ITI period in each condition. Inset shows grand average (mean) pupil size during ITI under the two social conditions. Error bars are 95% confidence interval across trials. (**) indicates p<0.01 and (***) shows p<0.001. In the interest of clarity, signals were smoothed using an averaging filter.

The online version of this article includes the following figure supplement(s) for figure 2:

**Figure supplement 1.** Pupil size correlates with participant's own confidence in the isolated condition.

**Figure supplement 2.** Time series analysis of pupil size during inter-trial interval.

conditions (*Figure 1c*). We observed that when participants were paired with a high (vs low) confidence partner, there was no significant difference in accuracy between the social conditions (p=0.92, p=0.75 for study 1 and 2 respectively, generalized linear mixed model [GLMM], see Supplementary materials for details of the analysis [*Table 1*], *Figure 1c* top-left panel); confidence, however, was significantly higher (p<0.001 for both studies, *Table 1*, *Figure 1c* middle panel) and RTs were significantly faster (p<0.001 for both, *Table 1*, *Figure 1c* bottom panel) in the HCA vs LCA.

This pattern of dissociations of speed and confidence from accuracy is non-trivial because the expectations of the standard sequential sampling models would be that a change in confidence should be reflected in change in accuracy (*Pouget et al., 2016*; *Sanders et al., 2016*). Many alternative mechanistic explanations are, in principle, possible. The rich literature on sequential sampling models in the random-dot paradigm permit articulating the components of such intuitive explanations as distinct computational models and comparing them by formal model comparison (see further below).

In order to assess the impact of social context on the participants' level of subjective uncertainty and rule out two important alternative explanations of confidence matching, we next examined the pupil data. Several studies have recently established a link between state of uncertainty and baseline (i.e. non-luminance mediated) variations in pupil size (*Bang et al., 2017*; *Wei and Wang, 2015*; *Nassar et al., 2012*; *Eldar et al., 2013*; *Murphy et al., 2014*; *Urai et al., 2017*). If the impact of social context on confidence were truly reflective of a similar change in the participant's belief about uncertainty, then we would expect the smaller pupil size when paired with high (HCA) vs low confidence agent (LCA) indicating lower subjective uncertainty. Alternatively, if confidence matching were principally due to pure imitation (*Rendell et al., 2011*; *Iacoboni, 2009*) or due to some form of social obligation in agreeing with others (e.g. normative conformity [*Stallen and Sanfey, 2015*]) without any change in belief, we would expect the pupil size to remain unaffected by pairing condition under social context. We found that during the inter-trial interval (ITI), pupil size was larger in the blocks where participants

**Table 2.** Details of statistical results in pupil data (*Figure 2*).

|         | Response | Regressors | Estimate | SE    | CI            | t-Stat | p-Value | Total number |
|---------|----------|------------|----------|-------|---------------|--------|---------|--------------|
| Study 1 | Pupil    | Condition  | –0.038   | 0.011 | [–0.06 –0.01] | –3.30  | <0.001  | 8390         |
| Study 2 | Pupil    | Condition  | –0.066   | 0.015 | [–0.09 –0.04] | –4.37  | <0.001  | 5842         |

were paired with LCA (vs HCA) (*Figure 2*, GLMM analysis, p<0.01 and p<0.001 for study 1 and 2 respectively, see Supplementary materials for details of the analysis; *Table 2*). We have added a time series analysis that demonstrates the temporal encoding of experimental conditions in the pupil signal during ITI (see *Figure 2—figure supplement 2*). It is important to bear in mind that pupil dilation has been linked to other factors such as mental effort (*Lee and Daunizeau, 2021*), level of surprise (*Kloosterman et al., 2015*), and arousal level (*Murphy et al., 2014*) as well. These caveats notwithstanding, the patterns of pupil dilation within the time period of ITI that are demonstrated and replicated here, are consistent with the hypothesis that participants' subjective belief was shaped by interactions with differently confident partners. To support this conclusion further, we provide supplementary evidence linking the participant's own confidence to pupil size (*Figure 2—figure supplement 1*).

To arbitrate between alternative explanations and develop a neural hypothesis for the impact of social context on decision speed and confidence, we constructed a neural attractor model (*Wong and Wang, 2006*), a variant from the family of sequential sampling models of choice under uncertainty (*Bogacz et al., 2006*). Briefly, in this model, noisy sensory evidence was sequentially accumulated by two competing mechanisms (red and blue in *Figure 3a* left) that raced toward a common pre-defined decision boundary (*Figure 3a* right) while mutually inhibiting each other. Choice was made as soon as one mechanism hits the boundary. This model has accounted for numerous observations of perceptual and value-based decision-making behavior and their underlying neuronal substrates in human (*Hunt et al., 2012*) and non-human primate (*Wei and Wang, 2015*) brain. Following previous works (*Wei and Wang, 2015*; *Balsdon et al., 2020*; *Rolls et al., 2010*; *Atiya et al., 2019*) we defined model confidence as the time-averaged difference between the activity of the winning and losing accumulators (corresponding to the shaded gray area between the two accumulator traces in *Figure 3a* right, for the model simulation see *Figure 3—figure supplement 2*) during the period of stimulus presentation (from 0 to 500 ms). Importantly, this definition of confidence is consistent with recent findings that computations of confidence continue *after* a decision has been made as long as sensory evidence is available (*Ruff and Fehr, 2014*; *Balsdon et al., 2020*; *van Kempen et al., 2019*; *Moran et al., 2015*). We also demonstrate that our results do not depend on this specific formulation and also replicate with another alternative method (*Vickers, 1979*)(see *Figure 3—figure supplement 3*).

Earlier works that demonstrated the relationship between decision uncertainty and pupil-related, global arousal state in the brain (*Murphy et al., 2014*; *Urai et al., 2017*) guided our modeling hypothesis. We modeled the social context as a global, top-down additive input (*Figure 3a*; $W_x$) in the attractor model. This input drove both accumulator mechanisms equally and positively. The impact of this global top-down input is illustrated in *Figure 3a* right: with a positive top-down drive ($W_x$>0), the winner (thick blue) and the loser (thick red) traces both rise faster compared to zero top-down drive (dotted lines). The model's counterintuitive feature is that the surface area between the winning and losing accumulator is larger in the case of positive (dark gray shading) versus zero (light gray shading) top-down input. Model simulations show that when $0<W_x$, this difference in surface area leads to faster RTs and higher confidence but does not change accuracy because it does not affect the decision boundary. These simulation results are consistent with our behavioral findings comparing HCA vs LCA conditions (*Figure 1c*).

We formally compared our model to three alternative, plausible models of how social context may affect the decision process. Without loss of generality, we used data from study 2 to fit the model. The first model hypothesized that partner's confidence dynamically modulated the decision bound (*Balsdon et al., 2020*) (parameter $B$ in *Equation 21*). In this model, the partner's higher confidence reduced the threshold for what counted as adequate evidence, producing the faster RTs under HCA (*Figure 1*.c). The second model proposed that partner's confidence changed non-decision time (NDT) (*Stine et al., 2020*; *Equation 22*). Here, pairing with high confidence partner would not have any impact on perceptual processing but instead, non-specifically decrease RTs across all coherence levels without affecting accuracy. Finally, in the third model, the stimulus-independent perceptual gain (*Eldar*

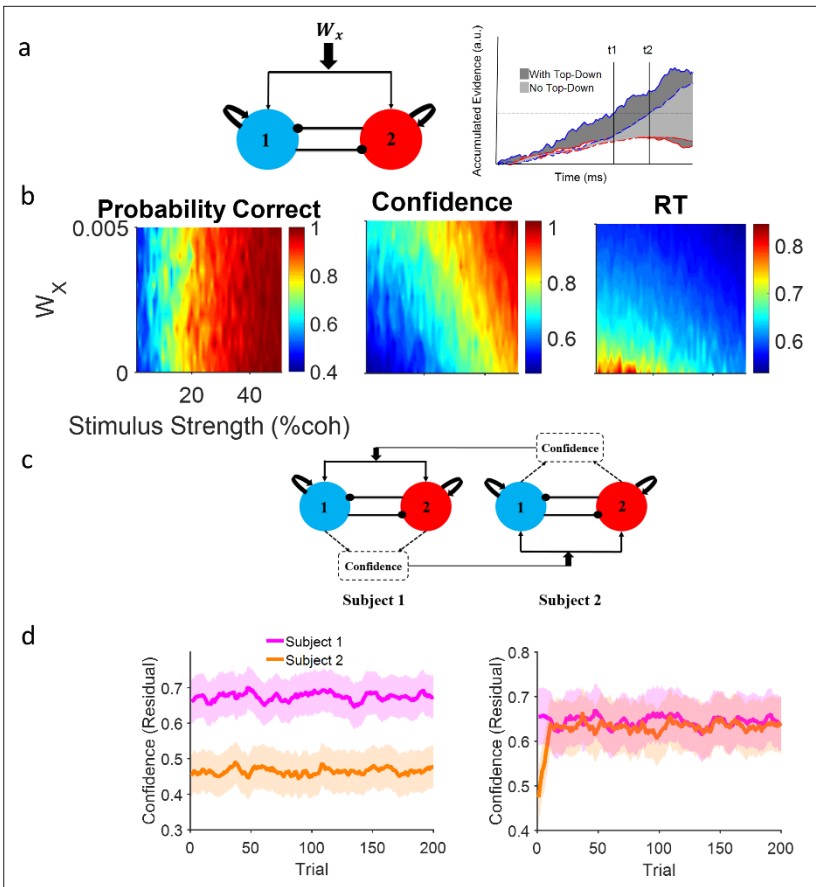

**Figure 3.** Neural attractor model. (**a**) Left: A common top-down ($W_x$) current drives both populations, each selective for a different choice alternative. Right: A schematic illustration of the impact of a positive top-down drive on accumulator dynamics. Confidence corresponds to the shaded area between winning (blue) and losing (red) accumulators. Solid lines and dark gray shade: positive top-down drive; dashed lines and light gray shade: zero top-down drive. With positive top-down current, the winner hits the bound earlier (*t1* vs *t2*) and the surface area between the competing accumulator traces is larger (dark vs light gray). (**b**) Systematic examination of the impact of $W_x$ on model behavior. Left panel: Accuracy does not depend on the top-down current but confidence (middle) and reaction time (RT) (right) change accordingly. Colors indicate different levels of top-down current. Each curve is the average of 10,000 simulations of the model given the top-down current. (**c**) Dynamic coupling in simulated dyadic interaction. Virtual dyads were constructed by feeding one model's confidence in previous trial to the other model as top-down drive and vice versa. (**d**) Left: Unconnected virtual dyad members ($W_x = 0$) simulate the isolated condition. Right: When the virtual dyad members are connected with top-down drive proportional to one another's confidence in previous trial, dyad members' confidence converge over time. In the isolated condition, confidence matching is not observed even though the pair receive the exact same sequence of stimuli. Shadowed areas of the confidence interval 95% resulted from 50 parallel simulations and curves were smoothed by an averaging filter for clearer illustration. The correlation with coherence has been removed from the confidence values via residual analysis (see *Figure 3—figure supplement 1* confidence values).

The online version of this article includes the following figure supplement(s) for figure 3:

**Figure supplement 1.** Confidence matching without removing the correlation with the shared stimulus coherence.

**Figure supplement 2.** The effect of top-down current on the attractor network.

**Figure supplement 3.** Model performance regarding different confidence representations.

**Figure supplement 4.** Model comparison.

**Figure supplement 5.** Model vs data.

**Figure supplement 6.** The speed of confidence matching.

**Figure supplement 7.** Model falsification.

**Figure supplement 8.** Model predictions for confidence matching are not sensitive to linearity assumptions.

et al., 2013; Li et al., 2018) parameter of input current (parameter $\mu_0$ in Equation 23) was modulated by partner confidence. Here, higher partner confidence increased the perceptual gain (as if increasing the volume of the radio) leading to increased confidence and decreased RT (Figure 1c) and would be consistent with the pupillometry results. In each model, in the social condition, the parameter of interest was linearly modulated by the confidence of the partner in the previous trial. Importantly, in Figure 1—figure supplement 3, we show that empirically, such trial-by-trial dependence is observed in confidence and RTs data in both study 1 and 2. Formal model comparison showed that our top-down additive current model was superior to all three alternatives (see Figure 3—figure supplement 4).

Having shown that a common top-down drive can qualitatively reproduce the impact of social context on speed-accuracy-confidence and quantitatively excel other alternatives in fitting the observed behavior, we then used the winning model to simulate our interactive social experiment virtually (Figure 3c). We simulated one decision maker with high confidence (subject 1 in Figure 3d) and another one with low confidence (subject 2). To simulate subject 1, we slightly increased the excitatory and the inhibitory weights. The opposite was done to simulate subject 2 (see Materials and methods for details). We then paired the two simulated agents by feeding the confidence of each virtual agent (from trial $t$–1) (Bang et al., 2017) as top-down input to the other virtual agent (in trial $t$).

Using this virtual social experiment, we simulated the dyadic exchanges of confidence in the course of our experiment and drew a prediction that could be directly tested against the empirical behavioral data. Without any fine-tuning of parameters or any other intervention, confidence matching emerged spontaneously when two virtual agents with very different confidence levels in isolated condition (Figure 3d left) were paired with each other as a dyad (Figure 3d right). Importantly, the model could be adapted to show different speed of matching as well (see Figure 3—figure supplement 6). However, for simplicity we presented the simplest case in the main text.

To identify the neural correlates of interpersonal alignment of belief about uncertainty, we note that previous works using non-invasive electrophysiological recordings in humans engaged in motion discrimination (Twomey et al., 2016; Stolk et al., 2013) have identified the signature, accumulate-to-bound neural activity characteristic of evidence accumulation in the sequential sampling process. Specifically, these findings show a centropareital positivity (CPP) component in the event-related potential that rises with sensory evidence accumulation across time. The exact correspondence between the neural CPP and elements of the sequential sampling process are not yet clear (O'Connell et al., 2018). For example, CPP could have resulted from the spatial superposition of the electrical activity of both accumulators or be the neural activity corresponding to the difference in accumulated evidence. These caveats notwithstanding, consistent with the previous literature, we found that in the isolated condition, our data replicated those earlier findings: Figure 4a shows a clear CPP event-related potential whose slope of rise was strongly modulated by motion coherence (GLMM, p<0.001 and p=0.01 for study 1 and 2 receptively, see Supplementary file 1d and Figure 4—figure supplement 2 for more details). Importantly, we have added the response-locked analysis of the CPP signals (see Figure 4—figure supplement 4). We do see that the response-locked CPP waveforms converge to one another for high vs low coherence trials at the moment of the response.

Our model hypothesized that under social condition, a top-down drive – determined by the partner's communicated confidence in the previous trial – would modulate the rate of evidence accumulation (Figure 3a). We tested if the CPP slope were larger within every given coherence bin when the participant was paired with an HCA (vs LCA). Indeed, the data demonstrated a larger slope of CPP rise under HCA vs LCA (Figure 4c, study 1 for the social condition p=0.15 but for the second study p<0.01, see Tables 3 and 4 for more details). These findings demonstrate that interpersonal alignment of confidence is associated with a modulation of neural evidence accumulation – as quantified by CPP – by the social exchange of information (also see Figure 4—figure supplement 3). It is important to note a caveat here before moving forward. These data show that both CPP and confidence are different between the HCA and LCA conditions. However, due to the nature of our experimental design, it would be premature to conclude from them that CPP contributes causally to the alignment of subjectively held beliefs or behaviorally expressed confidence. Put together with the behavioral confidence matching (Figure 1b) and the pupil data (Figure 2) our findings suggest that some such neural-social coupling could be the underlying basis for the construction of a shared belief about uncertainty.

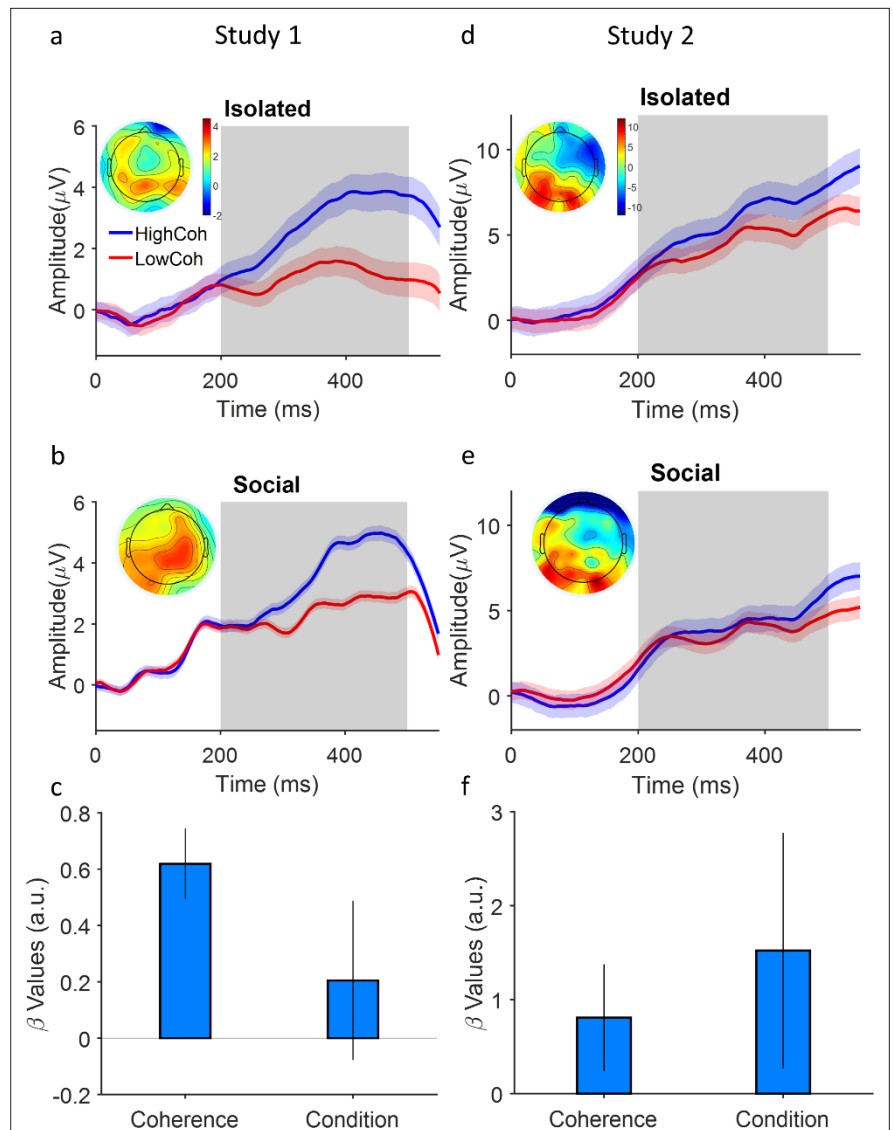

**Figure 4.** Coupling of neural evidence accumulation to social exchange of information. (**a**) Centroparietal positivity (CPP) component in the isolated condition: event-related potentials are time-locked to stimulus onset, binned for high and low levels of coherency (for study 1, low: 3.2%, 6.4%, 12.8%; high: 25.6% and 51.2%; for study 2 (**d**), low: 1.6%, 3.2%, 6.4%; high: 12.8%, 25.6%) and grand averaged across centropatrial electrodes (see Materials and methods). Inset shows the topographic distribution of the EEG signal averaged across the time window indicated by the gray area. (**b**) CPP under social condition. Conventions the same as panel (a). (**c**) A generalized linear mixed model (GLMM) model showed the significant relation of centroparietal signals to levels of coherency and social condition (high confidence agent [HCA] vs low confidence agent [LCA]). Error bars are 95% confidence interval over the model's coefficient estimates. Signals were smoothed by an averaging filter; shaded areas are SEM across trials.

The online version of this article includes the following figure supplement(s) for figure 4:

**Figure supplement 1.** Electrode placement in each study.

**Figure supplement 2.** Relation of EEG signals from centropartial area of the brain to coherence levels and social conditions.

**Figure supplement 3.** Simulated slope of the accumulator activity in our computational model in low confidence agent (LCA) and high confidence agent (HCA) conditions.

**Figure supplement 4.** Response-locked EEG signal separated for high vs low coherence levels.

**Figure supplement 5.** Power calculation (Monte Carlo simulation) for EEG slope effect (*Figure 4* in the main manuscript).

**Table 3.** Details of statistical results in EEG data (*Figure 4*).

|  | Response | Regressors | Estimate | SE | CI | t-Stat | p-Value | Total number |
|---|---|---|---|---|---|---|---|---|
|  |  | Coherency | 0.62 | 0.065 | [0.49. 074] | 9.64 | <0.001 | 6492 |
| Study 1 | EEG slope | Condition | 0.2 | 0.14 | [-0.07 0.49] | 1.42 | 0.15 | 6492 |
|  |  | Coherency | 0.8 | 0.29 | [0.24 1.37] | 2.8 | <0.01 | 5367 |
| Study 2 | EEG slope | Condition | 1.52 | 0.63 | [0.27 2.77] | 2.39 | 0.017 | 5367 |

## Discussion

We brought together two so-far-unrelated research directions: confidence in decision making under uncertainty and interpersonal alignment in communication. Our approach offers solutions to important current problems in each.

For decision science, we provide a model-based, theoretically grounded neural mechanism for going from individual, idiosyncratic representations of uncertainty (*Navajas et al., 2017*; *Fleming et al., 2010*) to socially transmitted confidence expressions (*Bahrami et al., 2010*; *Bang et al., 2017*) that are seamlessly shared and allow for successful cooperation. The social-to-neuronal coupling mechanism that we borrowed from the communication literature (*Hasson and Frith, 2016*; *Wheatley et al., 2019*) is crucial in this new understanding of the neuronal basis of relationship between subjectively private and socially shared uncertainty.

For communication science, by examining perceptual decision making under uncertainty in social context, we created a laboratory model in which the goal of communication was to arrive at a shared belief about uncertainty (rather than creating a look-up table for the meaning of actions [*Stolk et al., 2016*; *Silbert et al., 2014*; *Honey et al., 2012*]). In this way, we could employ the extensive theoretical, behavioral, and neurobiological body of knowledge in decision science (*Pouget et al., 2016*; *Adler and Ma, 2018*; *Aitchison et al., 2015*; *Hanks and Summerfield, 2017*; *Gold and Shadlen, 2007*; *Ruff and Fehr, 2014*; *Schall, 2019*; *Kiani et al., 2014*; *Kelly and O'Connell, 2013*; *Shadlen and Newsome, 2001*; *Resulaj et al., 2009*; *Sanders et al., 2016*; *Wei and Wang, 2015*; *Eldar et al., 2013*; *Urai et al., 2017*; *Yeung and Summerfield, 2012*; *Fleming and Daw, 2017*; *Kiani and Shadlen, 2009*) to construct a mechanistic neural hypothesis for interpersonal alignment.

Over the past few years, the efforts to understand the 'brain in interaction' have picked up momentum (*Wheatley et al., 2019*; *Frith and Frith, 1999*). A consensus emerging from these works is that, at a conceptual level, successful interpersonal alignment entails the mutual construction of a shared cognitive space between brains (*Stolk et al., 2015*; *Wheatley et al., 2019*; *Friston and Frith, 2015*). This would allow interacting brains to adjust their internal dynamics to converge on shared beliefs and meanings (*Hasson and Frith, 2016*; *Gallotti and Frith, 2013*). To identify the neurobiological substrates of such shared cognitive space, brain-to-brain interactions need to be described in terms of information flow, i.e., the impact that interacting partners have on one another's brain dynamics (*Wheatley et al., 2019*).

The evidence for such information flow has predominantly consisted of demonstrations of alignment of brain-to-brain activity (i.e. synchrony at macroscopic level, e.g. fMRI BOLD signal) when people process the same (simple or complex) sensory input (*Honey et al., 2012*; *Hasson et al., 2004*; *Breveglieri et al., 2014*; *Mukamel et al., 2005*; *Hasson and Honey, 2012*) or engage in complimentary communicative (*Silbert et al., 2014*) roles to achieve a common goal. More recently, dynamic coupling (rather than synchrony) has been suggested as a more general description of the nature of brain-to-brain interaction (*Hasson and Frith, 2016*). Going beyond the intuitive notions of synchrony and coupling, to our knowledge, no computational framework – grounded in the principles of neural

**Table 4.** Details of statistical results in EEG data (*Figure 4—figure supplement 2* top row).

|  | Response | Regressors | Estimate | SE | CI | t-Stat | p-Value | Total number |
|---|---|---|---|---|---|---|---|---|
| Study 1 | EEG slope | Coherency | 0.02 | 0.005 | [0.01 0.03] | 4.48 | <0.001 | 1523 |
| Study 2 | EEG slope | Coherency | 0.06 | 0.02 | [0.01 0.11] | 2.54 | <0.01 | 2822 |

computing – has been offered that could propose a plausible quantitative mechanism for these empirical observations of brain-to-brain coupling.

Combining four different methodologies, the work presented here undertook this task. Behaviorally, our participants engaged in social perceptual decision making under various levels of sensory and social uncertainty (*Bahrami et al., 2010*; *Bang et al., 2017*). Emergence of confidence matching (*Figure 1b*) showed that participants coordinated their decision confidence with their social partner. Pupil data (*Figure 2*) suggested that participant's belief about uncertainty was indeed shaped by the social coordination. A dissociation (*Figure 1c*) of decision speed and confidence from accuracy was reported that depended on the social context. This trade-off, as well as the emergence of confidence matching, was successfully captured by a neural attractor model (*Figure 3*) in which two competing neural populations of evidence accumulators – each tuned to one choice alternative – were driven by a common top-down drive determined by social information. This model drew predictions for behavior (*Figure 3d*) and neuronal activity (*Figure 4*, *Figure 4—figure supplements 1–5*) that were born out by the data. Social exchange of information modulated the neural signature of evidence accumulation in the parietal cortex.

Although numerous previous works have employed sequential sampling models to explain choice confidence, the overwhelming majority (*Pouget et al., 2016*; *Aitchison et al., 2015*; *Hanks and Summerfield, 2017*; *Gold and Shadlen, 2007*; *Ruff and Fehr, 2014*; *Schall, 2019*; *Kiani et al., 2014*; *Sanders et al., 2016*; *Kiani and Shadlen, 2009*; *Krajbich and Rangel, 2011*) have opted for the drift diffusion family of models. Neural attractor models have so far been rarely used to understand confidence (*Rolls et al., 2010*; *Atiya et al., 2019*; *Wang, 2002*). Our attractor model is a reduced version (*Wong and Wang, 2006*) of the original biophysical neural circuit model for motion discrimination (*Wang, 2002*). The specific affordances of attractor models allowed us to implement social context as a sustained, tonic top-down feedback to both accumulator mechanisms. More importantly, we were able to simulate social interactive decision making by virtually pairing any given two instances of the model (one for each member of a dyad) with each other: the confidence produced by each in a given trial served as top-down drive for the other in the next trial. Remarkably, a shared cognitive space about uncertainty (i.e. confidence matching) emerged spontaneously from this simulated pairing without us having to tweak any model parameters.

At a conceptual level, deconstructing the social communication of confidence into a comprehension and a production process (*Silbert et al., 2014*) is helpful. Comprehension process refers to how socially communicated confidence is incorporated in the recipient brain and affects their decision making. Production process refers to how the recipient's own decision confidence is constructed to be, in turn, socially expressed. It is tempting to attribute the CPP neural activity in the parietal cortex to the production process. Comprehension process, in turn, could be the top-down feedback from prefrontal brain areas previously implicated in confidence and metacognition (*Fleming et al., 2010*; *Fleming and Daw, 2017*; *De Martino et al., 2017*) to the parietal cortex. However, we believe that our neural attractor model in particular and the empirical findings do not lend themselves easily to this conceptual simplification. For example, the evidence accumulation process can be a part of the production (because confidence emerges from the integrated difference between accumulators) as well as the comprehension process (because the rate of accumulation is modulated by the received social information). As useful as it is, the comprehension/production dichotomy's limited scope should be recognized. Instead, armed with the quantitative framework of neural attractor models (for each individual) and interactive virtual pairing (to simulate dyads), future studies can now go beyond the comprehension/production dichotomy and examine the neuronal basis of interpersonal alignment with a model that have a strong footing in biophysical realities of neural computation.

Several limitations apply to our study. We chose different sets of coherence levels for the discovery (experiment 1) and replication (experiment 2). This choice was made deliberately. In experiment 1 we included a very high coherence (51%) level to optimize the experimental design for demonstrating the CPP component in the EEG signal. In experiment 2, we employed peri-threshold coherence levels in order to focus on behavior around the perceptual threshold to strengthen the model fitting and model comparison. This trade-off created some marginal differences in the observed effect sizes in the neural data across the two studies. The general findings were in good agreement.

The main strength of our work was to put together many ingredients (behavioral data, pupil and EEG signals, computational analysis) to build a picture of how the confidence of a partner, in the

context of joint decision making, would influence our own decision process and confidence evaluations. Many of the effects that we describe here are well described already in the literature but putting them all together in a coherent framework remains a challenge. For example, our study did not directly examine neural alignment between interaction partners. We measured the EEG signal one participant at a time. The participant interacted with an alleged (experimenter-controlled) partner in any given trial. Our experimental design, however, permitted strict experimental control and allowed us to examine the participants' social behavior (i.e. choices and confidence), pupil response, and brain dynamics as they achieved interpersonal alignment with the partner. Moreover, while the hypotheses raised by our neural attractor model did examine the nature of brain dynamics involved in evidence accumulation under social context, testing these hypotheses did not require hyper-scanning of two participants at the same time. We look forward to future studies that use the behavioral and computational paradigm described here to examine brain-to-brain neural alignment using hyper-scanning.

We have interpreted our findings to indicate that social information, i.e., partner's confidence, impacts the participants' beliefs about uncertainty. It is important to underscore here that, similar to real life, there are other sources of uncertainty in our experimental setup that could affect the participants' belief. For example, under joint conditions, the group choice is determined through the comparison of the choices and confidences of the partners. As a result, the participant has a more complex task of matching their response not only with their perceptual experience but also coordinating it with the partner to achieve the best possible outcome. For the same reason, there is greater outcome uncertainty under joint vs individual conditions. Of course, these other sources of uncertainty are conceptually related to communicated confidence, but our experimental design aimed to remove them, as much as possible, by comparing the impact of social information under high vs low confidence of the partner.

Our study brings together questions from two distinct fields of neuroscience: perceptual decision making and social neuroscience. Each of these two fields have their own traditions and practical common sense. Typically, studies in perceptual decision making employ a small number of extensively trained participants (approximately 6–10 individuals). Social neuroscience studies, on the other hand, recruit larger samples (often more than 20 participants) without extensive training protocols. We therefore needed to strike a balance in this trade-off between number of participants and number of data points (e.g. trials) obtained from each participant. Note, for example, that each of our participants underwent around 4000 training trials. Importantly, our initial study ($N$=12) yielded robust results that showed the hypothesized effects nearly completely, supporting the adequacy of our power estimate. However, we decided to replicate the findings in a new sample with $N$=15 participants to enhance the reliability of our findings and examine our hypothesis in a stringent discovery-replication design. In *Figure 4—figure supplement 5*, we provide the results of a power analysis that we applied on the data from study 1 (i.e. the discovery phase). These results demonstrate that the sample size of study 2 (i.e. replication) was adequate when conditioned on the results from study 1.

Finally, one natural limitation of our experimental setup is that the situation being studied is very specific to the design choices made by the experimenters. These choices were made in order to operationalize the problem of social interaction within the psychophysics laboratory. For example, the joint decisions were not an agreement between partners (*Bahrami et al., 2010*; *Bahrami et al., 2012*). Instead, following a number of previous works (*Bang et al., 2017*; *Bang et al., 2020*), joint decisions were automatically assigned to the most confident choice. In addition, partner's confidence and choice were random variables drawn from a distribution prespecified by the experimenter and therefore, by design, unresponsive to the participant's behavior. In this sense, one may argue that the interaction partner's behavior was not 'natural' since they did not react to the participant's confidence communications (note however that the partner's response times and accuracy were not entirely random but matched carefully to the participant's behavior prerecorded in the individual session). How much of the findings are specific to these experimental setting and whether the behavior observed here would transfer to other real-life settings is an open question. For example, it is plausible that participants may show some behavioral reaction to the response time variations since there is some evidence indicating that for binary choices like here, response times also systematically communicate uncertainty to others (*Patel et al., 2012*). Future studies could examine the degree to which the results might be paradigm-specific.

## Materials and methods

### Participants

A total of 27 participants (12 in experiment 1 and 15 in experiment 2; 10 females; average age: 24 years; all naïve to the purpose of the experiment) were recruited for a two-session experiment – isolated and social session. All subjects reported normal or corrected-to-normal vision. The participants did several training sessions in order to become familiar with the procedure and reach a consistent pre-defined level of sensitivity (see Materials and methods for more details).

### Recruitment

Participants volunteered to take part in the experiment in return for course credit for study 1. For study 2, a payment of 80,000 Toman equivalent to 2.5€ per session was made to each participant. On the experiment day, participants were first given the task instructions. Written informed consent was then obtained. The experiments were approved by the local Ethics Committee at Shaheed Rajaei University's Department of computer engineering.

### Task design

In the isolated session, each trial started with a red fixation point in the center of the screen (diameter 0.3°). Having fixated for 300 ms (in study 1, for a few subjects with eye monitoring difficulty this period shortened), two choice-target points appeared at 10° eccentricity corresponding to the two possible motion directions (left and right) (*Figure 1*). After a short random delay (200–500 ms, truncated exponential distribution), a dynamic RDM stimulus was displayed for 500 ms in a virtual aperture (5° diameter) centered on the initial fixation point. These motion stimuli have been described in detail elsewhere (*Shadlen and Newsome, 2001*). At the end of the motion stimulus a response panel (see *Figure 1a*) was displayed on the screen. This response panel consisted of a horizontal line extending from left to the right end of the display, centered on the fixation cross. On each side of the horizontal line, six vertical rectangles were displayed side by side (*Figure 1a*) corresponding to six confidence levels for each decision alternative. The participants reported the direction of the RDM stimulus and simultaneously expressed their decision and confidence using the mouse.

The rectangles on the right and left of the midpoint corresponded to the right and left choices, respectively. By clicking on the rectangles further the midpoint participants indicated higher confidence. In this way, participant indicated their confidence and choice simultaneously (*Kiani et al., 2014*; *Mahmoodi et al., 2015*) For experiment 1, response time was defined as the moment that the marker deviated (more than one pixel) from the center of the screen. However, in order to rule out the effect of unintentional movements, for the second study we increased this threshold to one degree of visual angle. The participants were informed about their accuracy by a visual feedback presented in the center of the screen for 1 s (correct or wrong).

In the social session, the participants were told they were paired with an anonymous partner. In fact, they were paired with a CGP tailored to the participant's own behavior in their isolated session. The participants did not know about this arrangement. Stimulus presentation and private response phase were identical to the isolated session. After the private response, the participants were presented with a social panel right (*Figure 1*). In this panel, the participant's own response (choice and confidence) were presented together with that of their partner for 1 s. The participant and the partner responses were color-coded (white for participants; yellow for partners). Joint decision was determined by the choice of the more confident person and displayed in green. Then, three distinct color-coded feedbacks were provided.

In both isolated and social sessions, the participants were seated in an adjustable chair in a semi-dark room with chin and forehead supported in front of a CRT display monitor (first study: 17 inches; PF790; refresh rate, 85 Hz; screen 164 resolution, 1024×768; viewing distance, 57 cm, second study: 21 inches; Asus VG248; refresh rate, 75 Hz; screen resolution, 1024×768; viewing distance, 60 cm). All the code was written in PsychToolbox (*Brainard, 1997*; *Kleiner et al., 2007*; *Pelli, 1997*).

### Training procedure

Each participant went through several training sessions (on average 4) to be trained on RDM task. They first trained in a response-free (i.e. RT) version of the RDM task in which motion stimulus was discontinued as soon as the participant responded. They were told to decide about the motion direction of

dots as fast and accurately as possible (*Kiani et al., 2014*). Once they reached a *stable* level of accuracy and RT, they proceeded to the main experiment. Before participating in the main experiment, they performed another 20–50 trials of warm-up. Here, the stimulus duration was fixed and responses included confidence report. For the social sessions, participants were told that in every block of 200 trials, they would be paired with a different person, seated in another room, with whom they would collaborate. They were also instructed about the joint decision scheme and were reminded that the objective in the social task was to maximize collective accuracy. Data from training and warm-up trials were included in the main analysis.

## Procedure

Each participant performed both the isolated and the social task. In the isolated session, they did one block containing 200 trials. Acquired data were employed to construct four computer partners for the first study and two partners for the second study. We used the procedure introduced in a previous works to generate CGPs (*Bang et al., 2017*; *Bang et al., 2022*). In the first study, the four partners were distinguished by their level of average accuracy and overall confidence: HAHC, HALC, LAHC, and finally LALC. For the second study partners only differed in confidence: HCA and LCA. Each participant performed one block of 200 trials for each of the paired partners – 800 overall for study 1 and 400 overall for study 2.

In the social session, participants were told to try to maximize the joint decision success (*Bang et al., 2017*). They were told that their payment bonus depended on by their joint accuracy (*Bang et al., 2020*). While performing the behavioral task, EEG signals and pupil data were also recorded.

## Computer generated partner

In study 1, following *Bang et al., 2017*, four partners were generated for each participant tuned to the participant's own behavioral data in the isolated session. Briefly, we created four simulated partners by varying their mean accuracy (high or low) and mean confidence (high or low). First, in the isolated session, the participant's sensory noise ($\sigma$) and a set of thresholds that determined the distribution of their confidence responses were calculated (see Materials and methods also). Simulated partner's accuracy was either high ($0.3 \times \sigma$) or low ($1.2 \times \sigma$). Mean confidence of simulated partners were also set according to the participant's own data. For low confidence simulated partner, average confidence was set to the average of participant's confidence in the low coherence (3.2% and 6.4%) trials. For the high confidence simulated partners, mean confidence was set to the average confidence of the participant in the high coherence (25.6% and 51.2%) trials. RTs were chosen randomly by sampling from a uniform random distribution (from 0.5 to 2 s). Thus, in some trials the participant needed to wait for the partner's response.

Having thus determined the parameters of the simulated partners, we then generated the sequence of trial-by-trial responses of a given partner using the procedure introduced by *Bang et al., 2017*. To produce the trial-by-trial responses of a given partner, we first generated a sequence of coherence levels with given directions (+ for rightward and – for leftward directions). Then we created a sequence of random values (sensory evidence), drawn from a Gaussian distribution with mean of coherence levels and variance of σ (sensory noise). Then, via applying the set of thresholds taken from the participant's data in isolated condition, we mapped the sequence of random values into trial-by-trial responses to generate a partner with a given confidence mean. Finally, to simulate lapses of attention and response errors, we randomly selected a response (from a uniform distribution over 1–6) on 5% of the trials (see *Figure 1—figure supplement 1* for the accuracy and confidence of the generated partners).

For study 2, we used the same procedure as study 1 and simulated two partners. These partners' accuracy was similar to the participant but each had a different confidence means (high confidence and low confidence partners). Therefore, we kept the σ constant and only change the confidence. For low confidence simulated partner, average confidence was set to the average of participant's confidence in the low coherence (1,6%, 3.2%, and 6.4%) trials. For the high confidence simulated partners, mean confidence was set to the average confidence of the participant in the high coherence (12.8% and 25.6%) trials.

## Signal detection theory model for isolated sessions

In study 1 and 2, we simulated 4 and 2 artificial partners, respectively. We followed the procedure described by *Bang et al., 2017*. Briefly, working with the data from the isolated session, the sensory

noise ($\sigma$) and response thresholds ($\theta$) for each participant were calculated using a signal detection theory model. In this model, the level of sensory noise ($\sigma$) determines the participant's sensitivity and a set of 11 thresholds determines the participant's response distribution, which indicate both decision (via its sign) and confidence within the same distribution (see below).

On each trial, the sensory evidence, $x$, is sampled from a Gaussian distribution, $x \in N(s, \sigma^2)$. The mean, $s$, is the motion coherence level and is drawn uniformly from the set $s \in S = \{-0.512, -0.256, -0.128, -0.064, -0.032, 0.032, 0.064, 0.128, 0.256, 0.512\}$ (for the second study $S = \{-0.256, -0.128, -0.064, -0.032, -0.016, 0.016, 0.032, 0.064, 0.128, 0.256\}$). The sign of $s$ indicates the correct direction of motion (right = positive) and its absolute value indicates the motion coherency. The standard deviation, $\sigma$, describes the level of sensory noise and is the same for all stimuli. We assumed that the internal estimate of sensory evidence ($z$) is equal to the raw sensory evidence ($x$). If $z$ is largely positive, it denotes high probability of choosing right direction and vice versa for largely negative values.

To determine the participant's sensitivity and the response thresholds, first, we calculated the distribution of responses ($r$, ranging from $-6$ to $6$, where the participant's confidence was ($c = |r|$), and her decision was determined by the sign of $r$). *Equation 1* shows the response distribution.

$$p_i = \begin{cases} p\left(z \leq \theta_{-6}\right) & i = -6 \\ p\left(\theta_{i-1} < z \leq \theta_i\right) & -6 < i \leq -1 \; or \; 2 \leq i < 6 \\ p\left(\theta_{-1} < z \leq \theta_1\right) & i = 1 \\ p\left(z > \theta_5\right) & i = 6 \end{cases} \tag{1}$$

Using $\theta$ and $\sigma$, we mapped $z$ to participants response ($r$). We found thresholds $\theta_i$ over $S$ where $i = -6, -5, -4, -3, -2, -1, 1, 2, 3, 4, 5$ such that:

$$\sum_{j \leq i} pj = \frac{1}{10} \sum_{s \in S} \Phi\left(\frac{\theta i - s}{\sigma}\right) \tag{2}$$

where $\Phi$ is the Gaussian cumulative density function. For each stimulus, $s \in S$, the predicted response distribution, $p\left(r = i|s\right)$, calculated by $S3$:

$$p\left(r = i|s\right) = \begin{cases} \left(\frac{\theta_{-6} - s}{\sigma}\right) & i = -6 \\ \left(\frac{\theta_i - s}{\sigma}\right) - \left(\frac{\theta_{i-1} - s}{\sigma}\right) & -6 < i < 6 \\ 1 - \left(\frac{\theta_5 - s}{\sigma}\right) & i = 6 \end{cases} \tag{3}$$

From here, the model's accuracy could be calculated by $S4$:

$$a_{agent} = \frac{\sum_{s \in S.s>0} \sum_{i=1}^{6} p_{i.s} + \sum_{s \in S.s<0} \sum_{i=-6}^{-1} p_{i.s}}{10} \tag{4}$$

Given participant's accuracy, we could find a set of $\theta$ and $\sigma$.

## Confidence estimation

Once we had determined $\theta$ and $\sigma$, we could produce a confidence landscape with a specific mean. In order to generate one high confidence and another low confidence partner, we needed to alter mean confidence by modifying the $\theta$. There could be an infinite number of confidence distribution with the desired mean. We were interested in the maximum entropy distribution that satisfied two constraints: mean confidence should be specified, and the distribution must sum to 1. Using Lagrange multiplier ($\lambda$) the response distribution was calculated as:

$$p_i = \frac{e^{i\lambda}}{\sum_{j=1}^{6} e^{i\lambda}} \tag{5}$$

with $\lambda$ chosen by solving the constraint

$$c = \frac{\sum_{j=1}^{6} j e^{j\lambda}}{\sum_{j=1}^{6} e^{j\lambda}} \tag{6}$$

We transformed confidence distributions (1–6) to response distributions (−6 to −1 and 1–6) by assuming symmetry around 0. *Figure 1—figure supplement 1* shows the accuracy and confidence of generated agents.

## Computational model

We employed a previously described attractor network model (*Wong and Wang, 2006*) which is itself the reduced version of an earlier one (*Wang, 2002*) inspired by the mean field theory. The model consists of two units simulating the average firing rates of two neural populations involved in information accumulation during perceptual decisions (*Figure 3a*). When the network is given inputs proportional to stimulus coherence levels, a competition breaks out between two alternative units. This race would continue until firing rates of one of the two units reaches the high-firing-rate attractor state at which point the alternative favored by the unit is chosen. The details of this model have been comprehensively described elsewhere (*Wong and Wang, 2006*).

Each unit was selective to one choice (*Equations 7; 8*) and received an input as follows:

$$x_1 = J_{N11}S_1 - J_{N12}S_2 + I_0 + I_1 + I_{noise1} \tag{7}$$
$$x_2 = J_{N22}S_2 - J_{N21}S_1 + I_0 + I_2 + I_{noise2} \tag{8}$$

where $J_{N11}$ and $J_{N22}$ indicated the excitatory recurrent connection of each population and $J_{N12}$ and $J_{N21}$ showed the mutual inhibitory connection values. For the simulation in *Figure 3b* we set the recurrent connections to 0.3157 nA and inhibitory ones to 0.0646 nA. $I_0$ indicated the effective external input which was set to 32.55 nA. $I_{noise1}/I_{noise2}$ stood for the internal noise in each population unit. This zero mean Gaussian white noise was generated based on the time constant of 2 ms and standard deviation of 0.02 nA. $I_1/I_2$ indicated the input currents proportional to the motion coherence level such that:

$$I_1 = J_{A.ext}\mu_0\left(1 + \frac{c}{100}\right) \tag{9}$$

$$I_2 = J_{A.ext}\mu_0\left(1 - \frac{c}{100}\right) \tag{10}$$

where $J_{A.ext}$ was the average synaptic coupling from the external source and set to 0.0002243 (nA Hz⁻¹), $c$ was coherence level and $\mu_0$, a.k.a. perceptual gain, was the input value when the coherence was zero (set to 45.8 Hz).

$S_1$ and $S_2$ were variables representing the synaptic current of either population and were proportional to the number of active NMDA receptors. Whenever the main text refers to accumulated evidence, we refer to $S_1$ and $S_2$ variables. Dynamics of these variables were as follows:

$$\frac{dS_1}{dt} = -\frac{S_1}{\tau_s} + (1 - S_1)\gamma H(x_1) \tag{11}$$

$$\frac{dS_2}{dt} = -\frac{S_2}{\tau_s} + (1 - S_2)\gamma H(x_2) \tag{12}$$

where $\tau_s$, the NMDA receptor delay time constant, was set to 100 ms, $\gamma$ set to 0.641 and the time step, $dt$, was set to 0.5 ms. Dynamical *Equations 11; 12* were solved using forward Euler method (*Wong and Wang, 2006*). (*H*), the generated firing rates of either populations, was calculated by:

$$H(x) = \frac{ax - b}{1 - e^{-d(ax-b)}} \tag{13}$$

where $a$, $b$, and $d$ were set to 270 Hz nA⁻¹, 108 Hz, and 0.154 s, respectively. These constants indicated the input-output relationship of a neural population.

The model's choice in each trial was defined as the accumulated evidence of either population that first touched a threshold, and the decision time was defined as the time when the threshold was touched. Notably, the decision threshold was set to $S_{threshold} = 0.32$. Moreover, the confidence was defined as the area between two accumulators ($S_1$ and $S_2$ in *Equations 11; 12*), in the time span of 0–500 ms, which was defined as:

$$Confidence = \left| \int_0^{500} (S_1 - S_2) \; dt \right| \tag{14}$$

which was normalized by following logistic function (**Wei and Wang, 2015**):

$$Normalized \; Confidence = b_1 + \frac{a}{e^{(kConfidence - b_0)}} \tag{15}$$

where the values of $b_1$, $a$, $k$, and $b_0$ were set to 1.32, –0.99, 5.9, and 0.16 respectively for model on entire trials of subjects in isolated sessions; *confidence* is calculated in **Equation 14** in time period of [0–500]ms.

In line with previous studies, we calculated the absolute difference between accumulators (**Equation 14**; **Wei and Wang, 2015**; **Rolls et al., 2010**). In this formulation, confidence is calculated from model activity during the stimulus duration (**Atiya et al., 2019**). Notably, in our confidence definition, we integrated the accumulators' difference even when the winning accumulator hit the threshold (post-decision period) (**Balsdon et al., 2020**; **Navajas et al., 2016**; **Yu et al., 2015**). This formulation of confidence provided a successful fit to subjects' behaviors (**Figure 3—figure supplement 5**). To demonstrate that our key findings do not depend on this specific formulation, we implemented another alternative method (**Vickers, 1979**) and showed qualitatively similar results (**Figure 3—figure supplement 3**) are obtained.

We calibrated the model to the data from the isolated condition to identify the best fitting parameters that would describe the participants' behavior in isolation. In this procedure decision threshold, inhibitory and excitatory connections, NDT (set 0.27 s) and $\mu_0$ were considered as the model variables (see **Supplementary file 1h** for parameter values).

In order to explain the role of social context on participant's behavior, we added a new input current to the model. Importantly we kept all other parameters of the model identical to the best fit to the participants' behavior in the isolated situation:

$$x_1 = J_{N11}S_1 - J_{N12}S_2 + I_0 + I_1 + I_{noise1} + W_x \tag{16}$$

$$x_2 = J_{N22}S_2 - J_{N21}S_1 + I_0 + I_2 + I_{noise2} + W_x \tag{17}$$

In order to evaluate the effect of $W_x$ on the RT, accuracy, and confidence, we simulated the model while systematically varying the values of $W_x$ (**Figure 3b**).

Having established the qualitative relevance of $W_x$ in providing a computational hypothesis for the impact of social context, then we defined $W_x$ proportional to the confidence of partner as follows:

**Table 5.** Details of statistical results for the impact of previous trial (**Figure 1—figure supplement 3**).

|  | Response | Regressors | Estimate | SE | CI | t-Stat | p-Value | Total number |
|---|---|---|---|---|---|---|---|---|
|  | Accuracy (HC vs LC) | Coherency | 0.007 | 0.0006 | [0.006 0.008] | 11.58 | <0.001 | 9600 |
|  |  | Conf ($t$–1) | –0.0017 | 0.005 | [–0.01 0.01] | –0.28 | 0.77 | 9600 |
|  | Confidence (HC vs LC) | Coherency | 0.047 | 0.001 | [0.045, 0.049] | 54.7 | <0.001 | 9600 |
|  |  | Conf ($t$–1) | 0.32 | 0.008 | [0.3 0.33] | 38.31 | <0.001 | 9600 |
| Study 1 | RT (HC vs LC) | Coherency | –0.005 | 0.0001 | [–0.0048 0.0044] | –44.36 | <0.001 | 9600 |
|  |  | Conf ($t$–1) | –0.0055 | 0.001 | [–0.007 –0.003] | –5.44 | <0.001 | 9600 |
|  | Accuracy (HC vs LC) | Coherency | 0.02 | 0.002 | [0.02 0.024] | 13.23 | <0.001 | 6000 |
|  |  | Conf ($t$–1) | 0.003 | 0.008 | [–0.012 0.018] | 0.37 | 0.7 | 6000 |
|  | Confidence (HC vs LC) | Coherency | 0.1 | 0.002 | [0.097 0.0106] | 47.2 | <0.001 | 6000 |
|  |  | Conf ($t$–1) | 0.09 | 0.01 | [0.07 0.11] | 8.6 | <0.001 | 6000 |
| Study 2 | RT (HC vs LC) | Coherency | –0.009 | 0.0003 | [–0.001 –0.008] | –26.2 | <0.001 | 6000 |
|  |  | Condition | 0.005 | 0.001 | [0.001 0.008] | 2.98 | <0.01 | 6000 |

$$W_x = \alpha.C_{partner(t-1)} \tag{18}$$

where $t$ was the trial number. The model inputs were identical to isolated situation expect for the top-down current of $W_x$ which indicates the social input where $\alpha$ was a normalization factor (or coupling coefficient) and $C_{partner(t-1)}$ indicates the partner's confidence in the previous trial. Thus, we added a social input based on the linear combination of the partner's confidence in the previous trial. Importantly the model performance is not sensitive to linearity assumptions (see *Figure 3—figure supplement 8*). Notably, the behavioral effect reported in the main script is also evident respect to the confidence of the agent in the previous trial (*Figure 1—figure supplement 3* and *Table 5*).

For simulations reported in *Figure 3d*, we created high and low confident models by altering the inhibitory and excitatory connections of the original model. For the high confident model, excitatory and inhibitory connections were set to 0.3392 and 0.0699. For the low confident model excitatory and inhibitory connections were set to 0.3163 and 0.0652 respectively. For the simulation of social interaction (*Figure 4f*), we coupled two instances of the model using *Equation 20* with $\alpha$ set to –0.0008 and 0.005 for high confident and low confident models, respectively. We ran the parallel simulations 50 times and reported the average results.

In order to remove the effect of coherence levels from models' confidence, we measured the residuals of models' confidence after regressing out the impact of coherence. Using this simple regression model:

$$\text{Model Confidence} = \beta_0 + \beta_1 \text{Coh} + \epsilon \tag{19}$$

where *Coh* is the motion coherence level and $\epsilon$ is the error term, we removed the information explainable by motion coherence levels from confidence data as following. Confidence residuals were therefore:

$$\text{Confidence Residuals} = \beta_0 + \epsilon \tag{20}$$

All the simulations of model in the text – and parameters reported in the method – are related to the model calibrated on the collapsed data of all subjects ($n=3000$ for isolated sessions of study 2).

## Alternative formulations for confidence in the computational model

In our main model, confidence is formalized by *Equation 14*. We calculated the integral of difference between the losing and the winning accumulator during the stimulus presentation. This value would then be fed into a logistic function (*Equation 15*) to produce the final confidence reported by the model (*Figure 3b* middle panel). To demonstrate the generality of our findings, we used another alternative (but similar) formulation in the previous literature for confidence representation. In *Figure 3—figure supplement 3*, we compare the resulting 'raw' confidence values (i.e. confidence values before they are fed to *Equation 15*).

Alternative formulations for confidence are:

1. For comparison we plot our main formulation (*Equation 14*) in *Figure 3—figure supplement 3a*.
2. By calculating the difference between winning and losing accumulator at the END of stimulus duration (*Navajas et al., 2016*; *Figure 3—figure supplement 3b*, we call this End method).

Our simulations showed that our formulation (*Figure 3—figure supplement 3a*) shows an expected modulation to top-down currents. *Figure 3—figure supplement 3b* also shows a similar pattern which indicates our results are not different from End method. Therefore, our computational results could be generalized to different confidence representation methods.

## Model comparison

For model comparison, we used the fitted parameters from the isolated session (study 2 only without loss of generality). The model parameters for the isolated condition were extracted for each participants in their own respective isolated session ($n=3000$ across all participants). Then we compared all 'alternative' models with a 'single free parameter' to determine the model with the best account to behavioral data in social sessions ($n=6000$ across all participants). We considered three alternative models for the comparison. Note that in all models $a$ is the normalization factor and the free parameter.

## Bound model

We hypothesized that partner's confidence modulates the participant's decision boundary according to:

$$B = B_{Isolated} + aConf_{t-1} \tag{21}$$

$B$ determines the threshold applied on the solution of the *Equations 11; 12* (see Materials and methods). $B_{Isolated}$ denotes the threshold in the isolated model. In this model, in social condition the bound depends on the value of the agent's confidence in the previous trial. Note that the optimum value of *a*, normalization or coupling factor, is most likely to be negative since it generates lower RTs in social vs isolated situation.

## NDT model

We hypothesized that NDT would be modulated by confidence of agent in the previous trial. Here,

$$NDT = NDT_{Isolated} + aConf_{t-1} \tag{22}$$

$NDT_{Isolated}$ was the NDT fitted on the isolated data. Similarly, the optimum *a* was expected to be negative.

## Gain model

We hypothesized that social information modulated the perceptual gain defined as:

$$\mu_0 = \mu_{0_{Isolated}} + aConf_{t-1} \tag{23}$$

where $\mu_0$ denotes the input value of the model when motion coherence is zero (*Equations 9; 10*, Materials and methods) and $\mu_{0_{Isolated}}$ was calculated based on isolated data. If *a* is positive, then $\mu_0$ would be greater under social condition vs isolated condition, which in turn generates lower RTs and higher confidence.

In order to incorporate the accuracy, RT, and confidence in model comparison, we calculated the RT distribution of trials in each of the 12 confidence levels, 6 for left decision (−6 to −1) and 6 for right decision (1–6). The RT in each level was further divided into two categories (*Ratcliff and McKoon, 2008*) (less than 700 ms and larger than 700 ms). We tried to maximize the likelihood of behavioral RT distribution in each response level (confidence and choice) given the model structure and parameters. The probability matrix was defined as follows:

$$Pmat = \left[ p_i \left( RT < 700 \right), p_i \left( RT > 700 \right) \right] \qquad -6 \leq i \leq 6 \tag{24}$$

where *i* is confidence levels ranging from −6 to 6. Note, the probability was calculated based on all trials in our behavioral data set (6000 trials). The model's probability matrix was also calculated in a similar manner. Hence, we derived a probability matrix of 12 response levels and 2 RT bins. The likelihood function was defined as follows:

$$JointPmat = |Pmat_{Behave} - Pmat_{Model}| \tag{25}$$

$$Cost = \sum_{i=1}^{12} \sum_{j=1}^{2} JointPmat_{(i,j)} \tag{26}$$

Since we used similar parameters for the models (all models had one free parameter, *a*) we could directly compare cost values corresponding to each model. The model with the lowest cost is the preferred model; the parameters were found via MATLAB *fmincon* function. As is often the case, there was some variability across participants (see *Figure 3—figure supplement 4*). To strengthen the conclusions about model comparison, we also provide evidence from a model falsification exercise that we performed. We simulated the models between two different social conditions (HCA and LCA) to see which model could, in theory, follow the behavioral pattern (*Figure 1c*). Indeed, we attempted to numerically *falsify* the alternative models. *Figure 3—figure supplement 7* shows the alternative model fails to reproduce the effect observed in *Figure 1c*.

## Eye monitoring and pupilometery

In both studies, the eye movements were recorded by an EyeLink 1000 (SR- Research) device with a sampling rate of 1000 Hz which was controlled by a dedicated host PC. The device was set in a desktop and pupil-corneal reflection mode while data from the left eye was recorded. At the beginning of each block, for most subjects, the system was recalibrated and then validated by 9-point schema presented on the screen. One subject was showed a 3-point schema due to the repetitive calibration difficulty. Having reached a detection error of less than 0.5°, the participants were led to the main task. Acquired eye data for pupil size were used for further analysis. Data of one subject in the first study was removed from further analysis due to storage failure.

Pupil data were divided into separate epochs and data from ITI were selected for analysis. ITI interval was defined as the time between offset of trial (*t*) feedback screen and stimulus presentation of trial (*t*+1). Then, blinks and jitters were detected and removed using linear interpolation. Values of pupil size before and after the blink were used for this interpolation. Data was also mid-pass filtered using Butterworth filter (second order, [0.01, 6] Hz) (*van Kempen et al., 2019*). The pupil data was

**Table 6.** The rate of trial rejection of eye tracking (only data of social) and EEG data (visual inspection) per participant.

|  | Participants | Eye tracking rejection % (social) | EEG trial rejection % (visual) |
|---|---|---|---|
|  | 1 | 12.25 | 4.6 |
|  | 2 | 12.87 | 31.1 |
|  | 3 | 0.5 | 22.1 |
|  | 4 | 4 | 14.8 |
|  | 5 | 1.37 | 34.4 |
|  | 6 | 0 | 4.6 |
|  | 7 | 7.75 | 8.8 |
|  | 8 | 0.37 | 24.4 |
|  | 9 | 6.37 | 7.6 |
|  | 10 | 0 | 46 |
|  | 11 | 0.12 | NA |
| Study 1 (Discovery) | 12 | NA | NA |
|  | 1 | 0 | 4 |
|  | 2 | 1.25 | 1 |
|  | 3 | 5.75 | 8.5 |
|  | 4 | 0.5 | 3 |
|  | 5 | 1 | 16 |
|  | 6 | 1.5 | 2.5 |
|  | 7 | 0 | 0.5 |
|  | 8 | 1.5 | 9 |
|  | 9 | 0 | 2 |
|  | 10 | 1 | 4 |
|  | 11 | 1 | 7.5 |
|  | 12 | 0.5 | 0 |
|  | 13 | 0.75 | 10.5 |
|  | 14 | 2.5 | 12 |
| Study 2 (Replication) | 15 | 14.75 | 4.5 |

z-scored and then was baseline corrected by removing the average of signal in the period of [–1000 0] ms interval (before ITI onset). Importantly, trials with ITI >3 s were excluded from analysis (365 out of 8800 for study 1 and 128 out 6000 for study 2; also see *Table 6* and Selection criteria for data analysis in Supplementary materials).

## EEG signal recording and preprocessing

For the first study, a 32-channel eWave32 amplifier was used for recording which followed the 10–10 convention of electrode placement on the scalp (for the locations of the electrodes, see *Figure 4—figure supplement 1*; right mastoid as the reference). The amplifier, produced by ScienceBeam (http://www.sciencebeam.com/), provided a 1 K sampling rate (*Vafaei Shooshtari et al., 2019*). For the second study we used a 64-channel amplifier produced by LIV team (http://lliivv.com/en/) with 250 Hz sampling rate (see the electrode placement in *Figure 4—figure supplement 1*).

Raw data were analyzed using EEGLAB software (*Delorme and Makeig, 2004*). First, data were notch filtered in the range of 45–55 Hz in order to remove the line noise. Using an FIR filter in the range of 0.1–100 Hz, high-frequency noise was also removed from data. Artifacts were removed by visual inspection using information from independent component analysis. Noisy trials were also removed by avisual inspection. Noisy channels were interpolated using EEGLAB software. The signals were divided into distinct epochs aligned to stimulus presentation ranging from 100 ms pre-stimulus onset until 500 ms post-stimulus offset. After preprocessing, EEG data in the designated epochs that had higher (lower) values than 200 (–200) μV were excluded from analysis (see *Table 6* and Materials and methods for detailed data analysis) (*Kelly and O'Connell, 2013*). We used CP1, CP2, Cz, and Pz electrodes for further analysis. In the first study, EEG recording was not possible in two participants due to unresolvable impedance calibration problems in multiple channels.

## Relation of CPP to coherence and social condition

Activities of centroparietal area of the brain is shown to be modulated with coherence level. Here, we showed that CPP activities are statistically related to the coherence levels (*Figure 4—figure supplement 2*, top-row) in both studies. Furthermore, we tested how much this relationship is dependent to social condition (HCA, LCA, *Figure 4—figure supplement 2*, bottom-row). Our analysis showed that the slope (respect to coherence levels) is different in HCA vs LCA (also see *Table 6*). Notably, this effect is in line with our neural model prediction (see *Figure 4—figure supplement 3*, next section).

## Selection criteria for data analysis

The data included in both studies could be classified into three main categories: behavioral, eye tracking, and EEG. For the behavioral analysis, data from all participants were included. In study 1, eye tracking data from one participant was lost due to storage failure. For pupil analysis, we excluded the trials with ITI longer than 3 s (~4% of trials in study 1 and ~2% for study 2).

**Table 7.** Generalized linear mixed model (GLMM) including interaction terms (p-values are reported).

|  | Response | Coherence | Condition (LC vs HC) | Condition* coherence |
|---|---|---|---|---|
|  | Accuracy | p<0.001 | p=0.92 | p=0.96 |
|  | Confidence | p<0.001 | p<0.001 | p<0.001 |
|  | RT | p<0.001 | p<0.001 | p<0.05 |
|  | Pupil | p=0.43 | p=0.20 | p=0.31 |
| Study 1 | EEG slope | p<0.01 | p=0.15 | p=0.91 |
|  | Accuracy | p<0.001 | p=0.75 | p=0.87 |
|  | Confidence | p<0.001 | p<0.001 | p<0.001 |
|  | RT | p<0.001 | p<0.001 | p=0.34 |
|  | Pupil | p=0.35 | p=0.06 | p=0.17 |
| Study 2 | EEG slope | p=0.62 | p<0.05 | p=0.68 |

**Table 8.** Attractor model's parameters.

| Parameter | Parameter value | Reference, remarks |
|---|---|---|
| $J_{N,ii}$ | 0.3157 nA | Calibrated based on pool of isolated data, also fitted on individual subjects' data |
| $J_{N,ij}$ | 0.0646 nA | Calibrated based on pool of isolated data, also fitted on individual subjects' data |
| $\mu_0$ | 45.8 Hz | Calibrated based on pool of isolated data, also fitted on individual subjects' data |
| NDT | 0.27 s | Calibrated based on pool of isolated data, also fitted on individual subjects' data |
| Bound | 0.32 nA | Calibrated based on pool of isolated data, also fitted on individual subjects' data |
| $a$ (**Equation 15**) | –0.99 | Calibrated based on pool of isolated data, also fitted on individual subjects' data |
| $b_0$ (**Equation 15**) | 1.32 | Calibrated based on pool of isolated data, also fitted on individual subjects' data |
| $b_1$ (**Equation 15**) | –0.165 | Calibrated based on pool of isolated data, also fitted on individual subjects' data |
| $k$ (**Equation 15**) | 5.9 | Calibrated based on pool of isolated data, also fitted on individual subjects' data |
| $I_0$ | 0.3255 nA | From **Wang, 2002**; **Wong and Wang, 2006** |
| $J_{A,ext}$ | 0.00022 nA Hz$^{-1}$ | From **Wang, 2002**; **Wong and Wang, 2006** |
| $\tau_s$ | 0.1 s | From **Wang, 2002**; **Wong and Wang, 2006** |
| $dt$ | 0.0005 s | From **Wang, 2002**; **Wong and Wang, 2006** |
| $a$ (**Equation 13**) | 270 (V nC)$^{-1}$ | From **Wang, 2002**; **Wong and Wang, 2006** |
| $b$ (**Equation 13**) | 108 Hz | From **Wang, 2002**; **Wong and Wang, 2006** |
| $d$ (**Equation 13**) | 0.154 s | From **Wang, 2002**; **Wong and Wang, 2006** |
| $\gamma$ | 0.641 | From **Wang, 2002**; **Wong and Wang, 2006** |
| Noise_std | 0.025 | From **Wang, 2002**; **Wong and Wang, 2006** |
| I_noise | 0.02 | From **Wang, 2002**; **Wong and Wang, 2006** |

We also analyzed brain data of participants in both studies. For the ERP analysis, we excluded trials with an absolute amplitude greater than 200 microvolts (overall less than 1% for both trials) as this data was deemed as outlier. Moreover, noisy trials and ICA components (around 5% of components in study 2) were rejected by visual inspection. Noisy electrodes were also interpolated (~8% of electrodes in study 2); see *Table 6* for more details. In study 1, EEG data from two participants were lost due to a technical failure. All data (behavioral, eye tracking, and EEG) for study 2 were properly stored, saved, and made available at https://github.com/JimmyEsmaily/ConfMatch (copy archived at *Esmaily, 2023*; *MathWorks Inc, 2023*).

## Statistical analysis

For hypothesis testing, we employed a number of GLMM. Unless otherwise stated, in our mixed models, participant was considered as random intercept. Details of each model is described in *Tables 1–6* in the Supplementary materials. This approach enabled us to separate the effects of coherency and partner confidence. For RT and confidence, we assumed that the data is normality distributed. For the accuracy data we assumed the distribution is Poisson. We used a maximum likelihood method for fitting. All p-values reported in the text were drawn from the GLMM method, unless stated otherwise. For completeness, for each analysis we have added interaction terms as well (see *Tables 7 and 8*).

## Permutation test to confirm confidence matching

A key null hypothesis ($p(\vartheta)$ where $\vartheta$ is the measure of interest: confidence matching) that we ruled out was that confidence matching was forced by the experimental design limitations and, therefore, would be observed in any random pairing of participants within our joint decision making setup. To reject this hypothesis, we performed a permutation test following *Bang et al., 2017* (see their Supplementary Figure 3 for further details). For each participant and corresponding CGP pair, we defined $|c_1 - c_2|$ where $c_i$ is the average confidence of participant $i$ in a given pair. We then estimated the null distribution for this variable by randomly re-pairing the participant with other participants and computing the mean confidence matching for each such re-paired set (total number of sets 1000). In *Figure 1—figure supplement 2* (bottom row), the red line shows the empirically observed mean of confidence matching in our data. The null distribution is shown in black. Proportion of values from the null distribution that were less than the empirical mean was $P \sim 0$.

In addition, we defined an index for measuring the confidence matching (*Figure 1—figure supplement 2*, first row): $\Delta m = \left| C_{isolated(Subject)} - C_{agent} \right| - \left| C_{social(Subject)} - C_{agent} \right|$. The larger the $\Delta m$ the higher is the confidence matching. Although we did not observe a significant effect of $\Delta m$, we showed that this index is significantly different from zero in the HCA condition.

## Acknowledgements

JE and BB were supported by the European Research Council (ERC) under the European Union's Horizon 2020 research and innovation programme (819040 – acronym: rid-O). BB was supported by the NOMIS foundation and Templeton Religion Trust.

## Additional information

### Funding

| Funder | Grant reference number | Author |
|---|---|---|
| European Research Council | 819040 - acronym: rid-O | Jamal Esmaily |

 The funders had no role in study design, data collection and interpretation, or the decision to submit the work for publication.

### Author contributions

Jamal Esmaily, Conceptualization, Data curation, Software, Formal analysis, Validation, Visualization, Methodology, Writing – original draft, Writing – review and editing; Sajjad Zabbah, Conceptualization, Data curation, Software, Methodology, Writing – original draft, Writing – review and editing; Reza Ebrahimpour, Conceptualization, Software, Supervision, Funding acquisition, Validation, Investigation, Methodology, Project administration, Writing – review and editing; Bahador Bahrami, Conceptualization, Resources, Supervision, Funding acquisition, Validation, Investigation, Visualization, Writing – original draft, Project administration, Writing – review and editing

### Author ORCIDs

Jamal Esmaily ⓘ https://orcid.org/0000-0001-5529-6732
Reza Ebrahimpour ⓘ https://orcid.org/0000-0002-7013-8078
Bahador Bahrami ⓘ https://orcid.org/0000-0003-0802-5328

### Ethics

Human subjects: Both experiments were approved by the local ethics committee at Faculty of computer engineering at Shahid Rajeie and also Iran University in Tehran, Iran (ethics application approval date and/or number: 5769). Written informed consent was obtained from all participants. The consent form was in the local Farsi language and did not include a "consent to publish" because data were anonymised, individual identity information was completely removed from the them and

none of the experimental hypotheses involved the exact identification of the individual data from any participant. Participants received a fixed monetary compensation for their contribution.

### Decision letter and Author response
Decision letter https://doi.org/10.7554/eLife.83722.sa1
Author response https://doi.org/10.7554/eLife.83722.sa2

## Additional files

### Supplementary files
• MDAR checklist

• Supplementary file 1. This file contains supplementary tables that contains details of statistical analysis.

### Data availability
Data that supports the findings of the study can be found here: https://osf.io/v7fqz/.

The following dataset was generated:

| Author(s) | Year | Dataset title | Dataset URL | Database and Identifier |
|---|---|---|---|---|
| Bahrami B, Esmaily J | 2020 | Neurobiology of Confidence Matching | https://osf.io/v7fqz/ | Open Science Framework, v7fqz |

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
