## [Editor Report]

This important study examines how humans use information about the confidence of collaborators to guide their own perceptual decision making and confidence judgements. The study addresses this question with a combination of psychophysics, electrophysiological modeling, and computational modelling that provides a compelling validation of a computational framework that can be used to derive and test theory-based predictions about how collaborators use communication to align their confidence and thereby optimize their collective performance.

---

## [Decision Letter]

**Decision letter after peer review:**

Thank you for submitting your article "Interpersonal alignment of neural evidence accumulation to social exchange of confidence" for consideration by *eLife*. Your article has been reviewed by 2 peer reviewers, and the evaluation has been overseen by a Reviewing Editor and Michael Frank as the Senior Editor. The reviewers have opted to remain anonymous.

Essential revisions:

Both reviewers agree that your paper has significant merit and represents a potentially important advance for the field. They also do highlight a number of areas where the description of methods needs to be clarified as well as control analyses that would need to be conducted in order to verify the reported results. The reviewers also highlight several areas where the authors' claims should be tempered and/or discussed in more detail.

1) Both reviewers query the extent to which the paradigm was representative of real-world cooperative decisions. Here, the decision was not agreed upon but rather assigned to the most confident observer. Examination of the degree to which the results might be paradigm-specific is warranted. Relatedly, there are several areas where the claims of the authors would need to be tempered or else additional analyses provided. Reviewer 1 highlights that the authors imply that causal links between pupil diameter, CPP, and confidence have been demonstrated when this is not the case. Analyses establishing such links should be conducted or else the authors' conclusions should be amended.

2) Both reviewers indicate that additional control analyses should be conducted to verify the connectivity results.

3) Further analyses need to be conducted in order to establish that the reported ERP signal really is a CPP. Response-locked ERP waveforms should be shown and analysed in order to verify that the observed signal bears the known functional properties of the CPP.

4) Further detail is required regarding the methods used for analysing the pupil data and for calibrating the stimuli to allow meaningful pupillometry. In addition, Reviewer 2 suggests employing a timeseries-based statistical framework rather than relying on averaging over arbitrary timeframes.

5) Both reviewers point to insufficient detail being provided regarding the model fitting procedures to allow for replication. Were all parameter values fit by the model or were some of them fixed according to the authors' own criteria?

*Reviewer #1 (Recommendations for the authors):*

Abstract: The authors write that their model "spontaneously demonstrated the emergence of social alignment". I don't think that this is correct, social alignment was not spontaneous, as a key aspect was introduced specifically for this purpose (using the confidence of one agent as a top-down drive for the decision process of the other agent).

Introduction: I find that the general use of citations should be improved. Many references are only vaguely related to the content they are supposed to support, and there is a bias towards citing studies that have been published recently and in high-impact journals, at the expense of more relevant ones.

Results:

Please report the method and results of the debriefing questionnaire.

Figure 1b would be easier to understand if the x- and y- axes were swapped, as the participant's confidence is an outcome whereas the agent's confidence is controlled by the experimenter if I understand it correctly.

In Figure 1 supplement 3, the authors write "The results did not change compared to figure 1c indicating the previous trials confidence impacts the behaviors in the upcoming trial regardless of experiment conditions (HCA, LCA)". This is not clear to me. Did they compare the effects reported in this figure between the two experimental conditions?

The authors interpret the larger pupil size (in blocks where participants are paired with a low-confidence agent) as reflecting participants' lower confidence. The evidence supporting this interpretation is far from clear. As noted by the authors, there are several alternative possibilities (arousal for instance). The authors might present more direct evidence linking confidence to pupil size in their own data, e.g. by examining this relation on a trial-by-trial basis, within each block or after the main effect of block is removed from the variables.

(Top of page 8) The 'intuitive description of the impact of global top-down input' does not provide a clear intuition to the reader. When both traces rise faster, why is there an increase in confidence? why does the difference between the two traces also increase in this case?

In the empirical and simulated data, it would be important to test the interactions between factors (coherence and partner's confidence, isolated vs. social) for completeness. In addition, the full specification of the GLMMs should be reported in the methods.

In their modelling work, the authors consider several candidate models in which a specific parameter is affected by the confidence of the partner in the previous trial. Then Figure 3d suggests that in simulations of the 'best' model, confidence matching occurs within the first 5 to 10 trials. When the initial trials are discarded, can we still observe confidence matching? It would be also important to compare this very fast convergence to that occurring in the real data. As the confidence of the partner is mostly modulated across different blocks of trials, it is unclear whether there is really a trial-by-trial adjustment beyond the first few initial trials within each block.

The model comparison (figure 3 —figure supplement 4) indicates that statistical comparisons between different models were done by using 20 initial points for each model. I don't think that this is relevant. Model fits should be estimated for each participant, and the fit quality can be then compared across participants. Figure 3 – supplement 5 is also not very informative. Individual data and fits, in the format of Figure 1c would be more relevant to examine the quality of the data and model fits.

The authors write: "These findings are the first neurobiological demonstration of interpersonal alignment by coupling of neural evidence accumulation to social exchange of information". It is not clear that the CPP contributes to the alignment of confidence. The authors have shown that both CPP and confidence are different between the HCA and LCA conditions. They have not shown that CPP and confidence are actually connected, nor that the change in one variable mediates the change in the other.

The finding of a greater information flow from the prefrontal to centro-parietal cortex in the HCA condition is potentially interesting, but not so convincing in its current presentation. It would be helpful to run control analyses in order to ensure that a flow in the opposite direction is not as likely and that the result would also not be present with a different region of interest instead of PFC. It would also be important to examine this same quantity in the non-social condition, in order to better qualify the results in the social condition.

In this section, it's also unclear how this increase in connectivity for HCA contributes to the CPP. Theoretical arguments and empirical data should be provided to address this.

Finally, I find the writing often unclear, difficult to follow and often trying to oversell the findings. I understand that this is subjective, but I suppose that simpler sentences and shorter expressions would make it easier for the reader (e.g. avoiding word bundles such as "socially observed emergent characteristics of confidence sharing", or "causally necessary neural substrate"). I find that clarity and concision would help the reader understand the true extent of the contribution of the study.

*Reviewer #2 (Recommendations for the authors):*

To address the weaknesses, it would help if the authors could:

1) Bolster sample sizes.

2) Provide a clearer definition of the types of uncertainties that are associated with communicated low confidence, and a discussion of which of these trigger the observed effects.

3) Describe the pupil analysis in much more detail, in particular how they calibrated the stimuli to allow interpretability of baseline signals and how they selected the specific traces in each trial's ITI so that they are not contaminated by stimuli.

4) Employ a more convincing time-series-based statistical framework for the analyses of the pupil data that does not rely on crude averaging over arbitrary timeframes, while correcting for multiple comparisons and autocorrelation.

5) Provide response-locked analyses of the EEG signals to establish that they correspond to the decision-linked CPPs as reported in the literature.

6) Provide much more information and justification for the selection of the signals that are interpreted as reflecting the top-down drive from the prefrontal cortex, and either also provide source-localization methods or any other type of evidence for the prefrontal origins of these signals. Also please display where in sensor space these signals are taken from (even that is missing).

7) Describe for every single parameter value that is reported whether it was produced by fitting the model and how exactly this was done, whether it was manually adjusted to produce a desired pattern, or whether it was set to a fixed value based on some clearly defined criteria. The aim is that others can replicate this work and use this model in the future, so this information is crucial!

8) Provide robustness analyses that show that the assumptions about linear modulation of parameters by confidence and the offset of the accumulation at the end of the stimulation period are justified.

9) Provide some more discussion about the unnatural properties of the fake interaction partners in this experiment, and to what degree this limits the interpretability of the findings and their generalizability to other contexts. Ideally, the authors would already show in a new sample and setup that the model can apply to real interactions, but that may be too much to ask for a single paper.

---

## [Author Response]

Essential revisions:Both reviewers agree that your paper has significant merit and represents a potentially important advance for the field. They also do highlight a number of areas where the description of methods needs to be clarified as well as control analyses that would need to be conducted in order to verify the reported results. The reviewers also highlight several areas where the authors' claims should be tempered and/or discussed in more detail.1) Both reviewers query the extent to which the paradigm was representative of real-world cooperative decisions. Here, the decision was not agreed upon but rather assigned to the most confident observer. Examination of the degree to which the results might be paradigm-specific is warranted. Relatedly, there are several areas where the claims of the authors would need to be tempered or else additional analyses provided. Reviewer 1 highlights that the authors imply that causal links between pupil diameter, CPP, and confidence have been demonstrated when this is not the case. Analyses establishing such links should be conducted or else the authors' conclusions should be amended.

In the discussion, the revised manuscript addresses the limitations of our design and discusses how our reduced version of interactive decision making differs from realworld joint decisions. Moreover, we have removed the implication of causal link between confidence, CPP and pupil diameter.

2) Both reviewers indicate that additional control analyses should be conducted to verify the connectivity results.

Thanks to reviewers’ comments, we have noted the caveat in our connectivity analysis. Having done a number of careful control analyses, we have decided to remove this analysis from the manuscript.

3) Further analyses need to be conducted in order to establish that the reported ERP signal really is a CPP. Response-locked ERP waveforms should be shown and analysed in order to verify that the observed signal bears the known functional properties of the CPP.

We have now addressed the concern about CPP signals. The revised results include response-locked ERP waveforms that bear the functional properties of CPP.

4) Further detail is required regarding the methods used for analysing the pupil data and for calibrating the stimuli to allow meaningful pupillometry. In addition, Reviewer 2 suggests employing a timeseries-based statistical framework rather than relying on averaging over arbitrary timeframes.

A number of modifications and additions have been made to the manuscript methods regarding the pupil data collection and analysis. We provide the necessary evidence for calibration of the pupil response to the stimulus luminance. In addition, we have reported the requested timeseries-based analysis for pupil data.

5) Both reviewers point to insufficient detail being provided regarding the model fitting procedures to allow for replication. Were all parameter values fit by the model or were some of them fixed according to the authors' own criteria?

We have now provided the detailed procedures of model fitting and model falsification. This includes a dedicated table indicating the procedure used (as well as the citation justifying it) for determining the value of each of the model parameters.

Reviewer #1 (Recommendations for the authors):Abstract: The authors write that their model "spontaneously demonstrated the emergence of social alignment". I don't think that this is correct, social alignment was not spontaneous, as a key aspect was introduced specifically for this purpose (using the confidence of one agent as a top-down drive for the decision process of the other agent).

We have now modified the abstract and removed the word “spontaneously”. In Lines 51-52 of the manuscript, the abstract now reads:

“An attractor neural network model incorporating social information as top-down additive input captured the observed behaviour and demonstrated the emergence of social alignment in virtual dyadic simulations.”

Introduction: I find that the general use of citations should be improved. Many references are only vaguely related to the content they are supposed to support, and there is a bias towards citing studies that have been published recently and in high-impact journals, at the expense of more relevant ones.

We have taken the reviewer’s opinion into account and revised our introduction. In doing so we have tried to trim and refine the selection of the cited papers and put more weight on earlier evidence that goes beyond the past few recent years. As indicated by the comment here, we tried to strike a tradeoff between a wider historical bracket and a narrower, more direct thematic relevance.

Results:Please report the method and results of the debriefing questionnaire.

The summary of the debriefing results is now included in as a supplementary figure (Figure 1—figure supplement 4). Importantly, none of the participants suspect their partner is a computer agent.

Figure 1b would be easier to understand if the x- and y- axes were swapped, as the participant's confidence is an outcome whereas the agent's confidence is controlled by the experimenter if I understand it correctly.

We have now swapped the axes in Figure 1b of the revised manuscript.

In Figure 1 supplement 3, the authors write "The results did not change compared to figure 1c indicating the previous trials confidence impacts the behaviors in the upcoming trial regardless of experiment conditions (HCA, LCA)". This is not clear to me. Did they compare the effects reported in this figure between the two experimental conditions?

We apologize for the confusion. In Figure 1c we illustrate the psychometric data (probability correct, confidence and RT) and experimental condition (high- or low confidence partner) is indicated by separate lines of different colours. In the corresponding analysis in the main text (lines 167-175) we compared each of the three aspects of behaviour across the two conditions. We found significant differences between the two conditions for confidence and RT. This analysis treated the condition as a block-design variable.

In the figure 1—figure supplement 3, we examine whether the participant’s behaviour (again probability correct, confidence and RT) in any given trial, is affected by the partner’s confidence in the immediate previous trial. As a result, in this analysis, the trial-by-trial variation in partner’s confidence is the key independent variable of interest. Here too, we the results showed a significant effect for confidence and RT but not for probability correct.

To be clear, the two analyses are two variations for testing the same hypothesis and they are not directly, formally compared with one another. The results of the two analyses are consistent with each other. We have now clarified this issue in the caption of Figure1-Supplement Figure 3.

The authors interpret the larger pupil size (in blocks where participants are paired with a low-confidence agent) as reflecting participants' lower confidence. The evidence supporting this interpretation is far from clear. As noted by the authors, there are several alternative possibilities (arousal for instance). The authors might present more direct evidence linking confidence to pupil size in their own data, e.g. by examining this relation on a trial-by-trial basis, within each block or after the main effect of block is removed from the variables.

To address this comment, we have now added a supplementary figure to Figure 2 where we provide supplementary evidence linking the participant’s own confidence to pupil size after partialing out the contribution of other factors. We employed a time-series analysis and examined the hypothesis that changes in pupil size are correlated with the participant’s own confidence in the isolated condition (also see figure 2—figure supplement 2). We found that confidence is encoded in the pupil size during inter-trial interval (ITI). Our findings are consistent with previous works (Urai et al., 2017) who showed that post-choice pupil signal varies with subjective confidence in perceptual decision making.

(Top of page 8) The 'intuitive description of the impact of global top-down input' does not provide a clear intuition to the reader. When both traces rise faster, why is there an increase in confidence? why does the difference between the two traces also increase in this case?

We have now clarified this point in the main text as follows (lines 255-265):

“We modeled the social context as a global, top-down additive input (Figure 3a; *Wx*) in the attractor model. This input drove both accumulator mechanisms equally and positively. The impact of this global top-down input is illustrated in Figure 3a right: with a positive top-down drive (Wx>0), the winner (thick blue) and the loser (thick red) traces both rise faster compared to zero top-down drive (dotted lines). The model’s counterintuitive feature is that the surface area between the winning and losing accumulator is larger in the case of positive (dark grey shading) versus zero (light grey shading) top-down input. Model simulations show that when *Wx* > 0, this difference in surface area leads to faster RTs and higher confidence but does not change accuracy because it does not affect the decision boundary. These simulation results are consistent with our behavioral findings comparing HCA vs LCA conditions (Figure 1c).”

In the empirical and simulated data, it would be important to test the interactions between factors (coherence and partner's confidence, isolated vs. social) for completeness. In addition, the full specification of the GLMMs should be reported in the methods.

We have updated this GLMM analysis and included the interaction terms. The results of this analysis are now reported in table Supplementary File 1g.

In their modelling work, the authors consider several candidate models in which a specific parameter is affected by the confidence of the partner in the previous trial. Then Figure 3d suggests that in simulations of the 'best' model, confidence matching occurs within the first 5 to 10 trials. When the initial trials are discarded, can we still observe confidence matching? It would be also important to compare this very fast convergence to that occurring in the real data. As the confidence of the partner is mostly modulated across different blocks of trials, it is unclear whether there is really a trial-by-trial adjustment beyond the first few initial trials within each block.

We are grateful to the reviewer for this idea. It is possible to answer this question both empirically and computationally. To empirically examine the speed of confidence matching, as requested here, in Figure 3-figure supplement 6, in the top box we have plotted the empirically observed timeline of confidence matching in the two studies. Here the absolute difference between the confidence of the agent and that of the subject is plotted against the trial number which indicates time. Observing the curves suggests that confidence matching starts quickly and then slows down as indicated by our simulations. The empirical data is, naturally, more noisy but the results do indicate that most of the matching happens at the very beginning. These empirical results come with the caveat that in our experiment, only one side of each pair, i.e., the participant behaviour was dynamically responsive and the partner’s responses were (by design) unaffected by the participant’s behaviour. Nonetheless, it is still possible to observe the trace of confidence matching in these data.

To answer the question at the level of the model, we note that our model in its basic form, does not flexibly adjust the speed of matching since we do not have any parameters to modify the amount of adjustment by the top down input. However, this could be achieved with a simple modification by adding a time constant to the top-down current. In this modified formulation, the amount of top-down current depends on the trial number such that *W_x_*(*t*) = *W_x_*_0_(*i*/(*i* + τ)) where *i* is the trial number, τ is the time constant and *W_x_*_0_ is the asymptotic value indicated by the dashed line in the panel. As can be observed in panels b-c, confidence matching is faster with lower values of τ.

We have now added the response to this comment to figure 3—figure supplement figure 6.

The model comparison (figure 3 —figure supplement 4) indicates that statistical comparisons between different models were done by using 20 initial points for each model. I don't think that this is relevant. Model fits should be estimated for each participant, and the fit quality can be then compared across participants. Figure 3 – supplement 5 is also not very informative. Individual data and fits, in the format of Figure 1c would be more relevant to examine the quality of the data and model fits.

We have now incorporated the requested changes into our fitting procedure and the reporting of the fitting results. In the revised manuscript, we report the following procedure:

First, we fitted the model to each single participant data, estimating a separate set of parameters for the Isolated Condition session and another one for the data from each of the Social Conditions i.e., LCA and HCA. The fit quality to different models was compared across participants as the reviewer suggested. As is often the case, there was some variability across participants (see the Pie chart figure 3—figure supplement 4).

To strengthen the conclusions about model comparison, we also provide evidence from a model falsification exercise that we performed.

Finally, with respect to the reviewer’s concern about visualizations of model fits to behavioral data, we have now modified figure 3—figure supplement 5 to the same conventions as Figure 1c.

To compare rival models, in addition to examining how well they fair in fitting to the empirical data, it is important and useful to examine if any of the models can be falsified in the light of the observed data. To do so, rather than fitting the models to the data from the social condition, here we first fit each candidate model to the data from the Isolated condition and extract the model parameters. Then we simulate the social condition (using the model-specific fitted parameters) to see if the simulation can qualitatively reproduce the pattern of empirical data (i.e., reaction time, accuracy, and confidence) observed empirically in the social condition. To recap, we compare our proposed model (top down input, TD) and three alternative models (Bound, NDT, Mu). For the TD model we have:

*TD_t_* = *TD_Isolated_* + *a*(*AgentConf_t_*_–1_) (R2)

Where *TD_isolated_*=0 and *AgentConf* is the confidence of the agent, *t* is the trial time and *a* is the coupling parameter. Similarly, we defined the model Bound, NDT and Mu, respectively, as follows:

*Bound_t_* = *Bound_Isolated_* + *a*(*AgentConf_t_*_–1_) (R3)

*NDT_t_* = *NDT_Isolated_* + *a*(*AgentConf_t_*_–1_) (R4)

*Mu_t_* = *Mu_Isolated_* + *a*(*AgentConf_t_*_–1_) (R5)

We simulated each of these models under different values of *a*. We used 0.003 and 0 for HCA (magenta) and LCA (orange) respectively. Similarly, we respectively used sets of (-0.15=*a_HCA*, 0=*a_LCA*), (-0.15, 0), (0.15, 0) as *a* for Bound, NDT, and Mu models. Then we simulated each model under these parameters 2000 times for each coherence level (form 0 to 25%). Then, we plotted the average of each model under each condition as shown Figure 3 – figure supplement 7.

Importantly, these simulations are not dependent on a specific parameter value. We have included the scripts for these model simulations here for readers to try different values for themselves https://github.com/Jimmy- 2016/ConfMatchEEG/tree/main/test_alternative_models. This repository is also listed in the manuscript.

Here, we also provide the intuition for why the alternative models fail to reproduce the observed behavioral effect:

Bound Model: Changing the decision bound would indeed change RT dramatically since the amount of evidence the model needs to decide would be different. However, a change in decision bound changes the accuracy as well. The higher the bound, the higher the accuracy. Most importantly, in our formulation, confidence does not depend on bound. We made this decision because there are many studies that have argued that confidence is related to post-decision evidence (Balsdon et al., 2020; Moran et al., 2015; Navajas et al., 2016; Yu et al., 2015). Therefore, changing the bound does not change the confidence.

NDT Model: Non-decision time is stimulus independent as can be seen in simulation results (Figure 3—figure supplement 7. C right panel). Therefore, we expect that increasing non-decision time would not have any effect on accuracy and confidence. This is what we observe in our simulations (Figure 3—figure supplement 7. C, left and middle). Importantly, this pattern was not what we have observed in behavioral data.

Mu Model: By increasing the Mu, the model becomes more sensitive to the evidence. This means it would accumulate the evidence more efficiently (faster and with more accuracy). Therefore by increasing the Mu, accuracy and confidence increase and RT decreases (Figure 3—figure supplement 7. D) which is inconsistent with our empirical observations.

The authors write: "These findings are the first neurobiological demonstration of interpersonal alignment by coupling of neural evidence accumulation to social exchange of information". It is not clear that the CPP contributes to the alignment of confidence. The authors have shown that both CPP and confidence are different between the HCA and LCA conditions. They have not shown that CPP and confidence are actually connected, nor that the change in one variable mediates the change in the other.

We have now followed the reviewer’s more cautious approach and modified the quoted section (line 501-510) as follows:

“These findings demonstrate that interpersonal alignment of confidence is associated with a modulation of neural evidence accumulation - as quantified by CPP - by the social exchange of information (also see figure 4-figure supplement 3). It is important to note a caveat here before moving forward. These data show that both CPP and confidence are different between the HCA and LCA conditions. However, due to the nature of our experimental design, it would be premature to conclude from them that CPP *contributes causally to* the alignment of subjectively held beliefs and/or behaviourally expressed confidence. Put together with the behavioral confidence matching (Figure 1b) and the pupil data (Figure 2) our findings suggest that some such neural-social coupling could be the underlying basis for the construction of a shared belief about uncertainty.”

The finding of a greater information flow from the prefrontal to centro-parietal cortex in the HCA condition is potentially interesting, but not so convincing in its current presentation. It would be helpful to run control analyses in order to ensure that a flow in the opposite direction is not as likely and that the result would also not be present with a different region of interest instead of PFC. It would also be important to examine this same quantity in the non-social condition, in order to better qualify the results in the social condition.

We are grateful to both reviewers for their notes of caution regarding the connectivity analysis. After having performed the control analyses suggested here, we have decided to remove this connectivity analysis from the manuscript. Most importantly, our analysis did not yield a convincing result when we swapped the PFC with a control region of interest. We have therefore come to the conclusion that it is best to leave the connectivity analysis out of the paper.

In this section, it's also unclear how this increase in connectivity for HCA contributes to the CPP. Theoretical arguments and empirical data should be provided to address this.

Following from removing the connectivity analysis, we have also removed the corresponding speculation about the role of increased connectivity under HCA condition.

Finally, I find the writing often unclear, difficult to follow and often trying to oversell the findings. I understand that this is subjective, but I suppose that simpler sentences and shorter expressions would make it easier for the reader (e.g. avoiding word bundles such as "socially observed emergent characteristics of confidence sharing", or "causally necessary neural substrate"). I find that clarity and concision would help the reader understand the true extent of the contribution of the study.

In the revised manuscript, we have taken the reviewer’s advice on board and attempted to use simpler and clearer language and avoid overly exaggerated claims.

Reviewer #2 (Recommendations for the authors):To address the weaknesses, it would help if the authors could:1) Bolster sample sizes.

Our study brings together questions from two distinct fields of neuroscience: perceptual decision making and social neuroscience. Each of these two fields have their own traditions and practical common sense. Typically, studies in perceptual decision making employ a small number of extensively trained participants (approximately 6 to 10 individuals). Social neuroscience studies, on the other hand, recruit larger samples (often more than 20 participants) without extensive training protocols. We therefore needed to strike a balance in this trade-off between number of participants and number of data points (e.g. trials) obtained from each participant. Note, for example, that each of our participants underwent around 4000 training trials. Strikingly, our initial study (N=12) yielded robust results that showed the hypothesized effects nearly completely, supporting the adequacy of our power estimate. However, we decided to replicate the findings because, like the reviewer, we believe in the importance of adequate sampling. We increased our sample size to N=15 participants to enhance the reliability of our findings. However, we acknowledge the limitation of generalizing to larger samples, which we have now discussed in our revised manuscript and included a cautionary note regarding further generalizations.

To complement our results and add a measure of their reliability, here we provide the results of a power analysis that we applied on the data from study 1 (i.e. the discovery phase). These results demonstrate that the sample size of study 2 (i.e. replication) was adequate when conditioned on the results from study 1. The results showed that N=13 would be an adequate sample size for 80% power for behavoural and eye-tracking measurements. Power analysis for the EEG measurements indicated that we needed N=17. Combining these power analyses. Our sample size of N=15 for Study 2 was therefore reasonably justified.

We have now added a section to the discussion (Lines 790-805) that communicates these issues as follows:

“Our study brings together questions from two distinct fields of neuroscience: perceptual decision making and social neuroscience. Each of these two fields have their own traditions and practical common sense. Typically, studies in perceptual decision making employ a small number of extensively trained participants (approximately 6 to 10 individuals). Social neuroscience studies, on the other hand, recruit larger samples (often more than 20 participants) without extensive training protocols. We therefore needed to strike a balance in this trade-off between number of participants and number of data points (e.g. trials) obtained from each participant. Note, for example, that each of our participants underwent around 4000 training trials. Importantly, our initial study (N=12) yielded robust results that showed the hypothesized effects nearly completely, supporting the adequacy of our power estimate.

However, we decided to replicate the findings in a new sample with N=15 participants to enhance the reliability of our findings and examine our hypothesis in a stringent discovery-replication design. In Figure 4-figure supplement 5, we provide the results of a power analysis that we applied on the data from study 1 (i.e. the discovery phase). These results demonstrate that the sample size of study 2 (i.e. replication) was adequate when conditioned on the results from study 1.”

We conducted Monte Carlo simulations to determine the sample size required to achieve sufficient statistical power (80%) (Szucs and Ioannidis, 2017). In these simulations, we utilized the data from study 1. Within each sample size (N, x-axis), we randomly selected N participants from our 12 participants in study 1. We employed the with-replacement sampling method. Subsequently, we applied the same GLMM model used in the main text to assess the dependency of EEG signal slopes on social conditions (HCA vs LCA). To obtain an accurate estimate, we repeated the random sampling process 1000 times for each given sample size (N). Consequently, for a given sample size, we performed 1000 statistical tests using these randomly generated datasets. The proportion of statistically significant tests among these 1000 tests represents the statistical power (y-axis). We gradually increased the sample size until achieving an 80% power threshold, as illustrated in the figure. The number indicated by the red circle on the x axis of this graph represents the designated sample size.

2) Provide a clearer definition of the types of uncertainties that are associated with communicated low confidence, and a discussion of which of these trigger the observed effects.

We appreciate the reviewer’s advice to remain cautious about the possible sources of uncertainty in our experiment. In the Discussion (lines 790-801) we have now added the following paragraph.

“We have interpreted our findings to indicate that social information, i.e. partner’s confidence, impacts the participants beliefs about uncertainty. It is important to underscore here that, similar to real life, there are other sources of uncertainty in our experimental setup that could affect the participants' belief. For example, under joint conditions, the group choice is determined through the comparison of the choices and confidences of the partners. As a result, the participant has a more complex task of matching their response not only with their perceptual experience but also coordinating it with the partner to achieve the best possible outcome. For the same reason, there is greater outcome uncertainty under joint vs individual conditions. Of course, these other sources of uncertainty are conceptually related to communicated confidence but our experimental design aimed to remove them, as much as possible, by comparing the impact of social information under high vs low confidence of the partner.”

In addition to the above, we would like to clarify one point here with specific respect to the comment. Note that the computer-generated partner’s accuracy was identical under high and low confidence. In addition, our behavioral findings did not show any difference in accuracy under HCA and LCA conditions. As a consequence, the argument that “the trial outcome is determined by the joint performance of both partners, which is normally reduced for low-confidence partners” is not valid because the low-confidence partner’s performance is identical to that of the high-confidence partner. It is possible, of course, that we have misunderstood the reviewer’s point here and we would be happy to discuss this further if necessary.

In Author response image 1 we provide the calibration plot for our eye tracking setup, describing the relationship between pupil size and display luminance. Luminance of the random dot motion stimuli (ie white dots on black background) was Cd/m^2^ and, importantly, identical across the two critical social conditions. We hope that this additional detail satisfies the reviewer’s concern. For the purpose of brevity, we have decided against adding this part to the manuscript and supplementary material.

**Author response image 1. sa2fig1:** Calibration plot for the experimental setup. Average pupil size (arbitrary units from eyelink device) is plotted against display luminance. The plot is obtained by presenting the participant with uniform full screen displays with 10 different luminance levels covering the entire range of the monitor RGB values (0 to 255) whose luminance was separately measured with a photometer. Each display lasted 10 seconds. Error bars are standard deviation between sessions.

Moreover, while the authors state that the traces were normalized to a value of 0 at the start of the ITI period, the data displayed in Figure 2 do not show this normalization but different non-zero values. Are these data not normalized, or was a different procedure used?

Finally, the authors analyze the pupil signal averaged across a wide temporal ITI interval that may contain stimulus-locked responses (there is not enough information in the manuscript to clearly determine which temporal interval was chosen and averaged across, and how it was made sure that this signal was not contaminated by stimulus effects).

3) Describe the pupil analysis in much more detail, in particular how they calibrated the stimuli to allow interpretability of baseline signals and how they selected the specific traces in each trial's ITI so that they are not contaminated by stimuli.

We have now added the following details to the Methods section in line 1106-1135.

“In both studies, the Eye movements were recorded by an EyeLink 1000 (SR- Research) device with a sampling rate of 1000Hz which was controlled by a dedicated host PC. The device was set in a desktop and pupil-corneal reflection mode while data from the left eye was recorded. At the beginning of each block, the system was recalibrated and then validated by 9-point schema presented on the screen. For one subject was, a 3-point schema was used due to repetitive calibration difficulty. Having reached a detection error of less than 0.5°, the participants proceeded to the main task. Acquired eye data for pupil size were used for further analysis. Data of one subject in the first study was removed from further analysis due to storage failure.

Pupil data were divided into separate epochs and data from Inter-Trials Interval (ITI) were selected for analysis. ITI interval was defined as the time between offset of trial (t) feedback screen and stimulus presentation of trial (t+1). Then, blinks and jitters were detected and removed using linear interpolation. Values of pupil size before and after the blink were used for this interpolation. Data was also mid-pass filtered using a Butterworth filter (second order, [0.01, 6] Hz)[50]. The pupil data was z-scored and then was baseline corrected by removing the average of signal in the period of [-1000 0] ms interval (before ITI onset). For the statistical analysis (GLMM) in Figure 2, we used the average of the pupil signal in the ITI period. Therefore, no pupil value is contaminated by the upcoming stimuli. Importantly, trials with ITI>3s were excluded from analysis (365 out of 8800 for study 1 and 128 out 6000 for study 2. Also see Supplementary File 1f and Selection criteria for data analysis in Supplementary Materials)”

4) Employ a more convincing time-series-based statistical framework for the analyses of the pupil data that does not rely on crude averaging over arbitrary timeframes, while correcting for multiple comparisons and autocorrelation.

In the revised manuscript, we have added a time series analysis that demonstrates the temporal encoding of experimental conditions in the pupil signal. We employed a general linear mixed model (GLMM) using the model Pupil(t) = b0 + b1*socialCondition

where social conditions are HCA=1 and LCA=2. The GLMM was applied to the time window from 0 to 2 seconds after the onset of ITI. We have plotted the time course of b1 (error bar = standard error obtained from the GLMM). We discretized time into nonoverlapping bins of 100ms width and averaged the data points falling within each bin.

This analysis is now added to the manuscript in figure 2—figure supplement 2.

5) Provide response-locked analyses of the EEG signals to establish that they correspond to the decision-linked CPPs as reported in the literature.

We are grateful to the reviewer for this helpful suggestion. We have now added the response-locked analysis of the CPP signals (see figure 4- supplement figure 4). As expected by the reviewer, we do see that the response-locked CPP waveforms converge to one another for high vs low coherence trials at the moment of the response. This pattern is more clearly seen in experiment 2. It is worth noting that previous studies that examined response-locked CPP employed reaction time (or long duration) tasks with variable stimulus duration. In our study, however, stimulus duration was fixed. Our data therefore provide a new addition to this literature confirming response locked CPP do not depend on termination of stimulus by response. We have now added the information to the manuscript.

6) Provide much more information and justification for the selection of the signals that are interpreted as reflecting the top-down drive from the prefrontal cortex, and either also provide source-localization methods or any other type of evidence for the prefrontal origins of these signals. Also please display where in sensor space these signals are taken from (even that is missing).

We are grateful to both reviewers for their notes of caution regarding the connectivity analysis. After having performed the control analyses suggested by reviewer 1 and the checks requested here, we have decided to remove this connectivity analysis from the manuscript.

7) Describe for every single parameter value that is reported whether it was produced by fitting the model and how exactly this was done, whether it was manually adjusted to produce a desired pattern, or whether it was set to a fixed value based on some clearly defined criteria. The aim is that others can replicate this work and use this model in the future, so this information is crucial!

In Supplementary material Supplementary File 1h, we have now described, for every single parameter that is reported, whether they adjusted (manually or fitted) or fixed. In each case, the relevant reference is also reported.

Please note that since we use the same model as (Wong, 2006), we kept most of the parameters the same as the original study. The following 5 parameters were modified

1. Decision threshold

2. Mu0 which is the gain

3. Recurrent connection (JN11 and JN22)

4. Inhibitory connection (JN12 and JN21) 5. Non-decision time.

Moreover, in order to map the difference between S1 and S2 activity to the confidence landscape (from 0 to 1), we used a mapping function as described in (Wei and Wang, 2015) and described as the following equation:

*Normalized Conf* = *b*1 + *a*/exp(*K* * *confdiff* – *b*0) (R5)

Where *confdiff* is the sum of the difference between losing and winning accumulators during the stimulus presentation (0-500) and *b_1_*, *a*, *K*, *b_0_* are free parameters. We have now reported this table and the accompanying explanations in the Supplementary material Supplementary File 1h.

8) Provide robustness analyses that show that the assumptions about linear modulation of parameters by confidence and the offset of the accumulation at the end of the stimulation period are justified.

We have now added the following further analysis and showed that our model is indeed not sensitive to linearity assumption (see figure 3—figure supplement 8).

9) Provide some more discussion about the unnatural properties of the fake interaction partners in this experiment, and to what degree this limits the interpretability of the findings and their generalizability to other contexts. Ideally, the authors would already show in a new sample and setup that the model can apply to real interactions, but that may be too much to ask for a single paper.

A similar concern was expressed by reviewer 1. We have now acknowledged this caveat in the discussion in line 485 to 504. The final paragraph of the discussion now reads as follows:

“Finally, one natural limitation of our experimental setup is that the situation being studied is very specific to the design choices made by the experimenters. These choices were made in order to operationalize the problem of social interaction within the psychophysics laboratory. For example, the joint decisions were not an agreement between partners (Bahrami et al., 2010, 2012). Instead, following a number of previous works (Bang et al., 2017, 2020) joint decisions were automatically assigned to the most confident choice. In addition, partner’s confidence and choice were random variables drawn from a distribution prespecified by the experimenter and therefore, by design, unresponsive to the participant’s behaviour. In this sense, one may argue that the interaction partner’s behaviour was not “natural” since they did not react to the participant's confidence communications (note however that the partner’s response times and accuracy were not entirely random but matched carefully to the participant’s behavior prerecorded in the individual session). How much of the findings are specific to these experimental setting and whether the behavior observed here would transfer to other real-life settings is an open question. For example, it is plausible that participants may show some behavioral reaction to the response time variations since there is some evidence indicating that for binary choices like here, response times also systematically communicate uncertainty to others (Patel et al., 2012). Future studies could examine the degree to which the results might be paradigm-specific.”

References cited in the rebuttal:

Bahrami, B., Olsen, K., Bang, D., Roepstorff, A., Rees, G., and Frith, C. (2012). What failure in collective decision-making tells us about metacognition. *Philosophical Transactions of the Royal Society B: Biological Sciences*, *367*(1594), 1350–1365. https://doi.org/10.1098/rstb.2011.0420

Bahrami, B., Olsen, K., Latham, P. E., Roepstorff, A., Rees, G., and Frith, C. D. (2010).

Optimally interacting minds. *Science*, *329*(5995), 1081–1085. https://doi.org/10.1126/science.1185718

Balsdon, T., Wyart, V., and Mamassian, P. (2020). Confidence controls perceptual evidence accumulation. *Nature Communications*, *11*(1753), 1–11. https://doi.org/10.1038/s41467-020-15561-w

Bang, D., Aitchison, L., Moran, R., Herce Castanon, S., Rafiee, B., Mahmoodi, A., Lau, J. Y. F., Latham, P. E., Bahrami, B., and Summerfield, C. (2017). Confidence matching in group decision-making. *Nature Human Behaviour*, *1*(6), 0117. https://doi.org/10.1038/s41562-017-0117

Bang, D., Ershadmanesh, S., Nili, H., and Fleming, S. M. (2020). Private–public mappings in human prefrontal cortex. *eLife*, *9*, 1–25. https://doi.org/10.7554/*eLife*.56477

Kelly, S. P., and O’Connell, R. G. (2013). Internal and External Influences on the Rate of Sensory Evidence Accumulation in the Human Brain. *Journal of Neuroscience*, *33*(50),

19434–19441. https://doi.org/10.1523/JNEUROSCI.3355-13.2013

Loughnane, G. M., Newman, D. P., Tamang, S., Kelly, S. P., and O’Connell, R. G. (2018). Antagonistic Interactions Between Microsaccades and Evidence Accumulation

Processes During Decision Formation. *The Journal of Neuroscience*, *38*(9), 2163–

2176. https://doi.org/10.1523/JNEUROSCI.2340-17.2018

Moran, R., Teodorescu, A. R., and Usher, M. (2015). Post choice information integration as a causal determinant of confidence: Novel data and a computational account. *Cognitive Psychology*, *78*, 99–147. https://doi.org/10.1016/j.cogpsych.2015.01.002

Navajas, J., Bahrami, B., and Latham, P. E. (2016). Post-decisional accounts of biases in confidence. *Current Opinion in Behavioral Sciences*, *11*, 55–60. https://doi.org/10.1016/j.cobeha.2016.05.005

O’Connell, R. G., Dockree, P. M., and Kelly, S. P. (2012). A supramodal accumulation-tobound signal that determines perceptual decisions in humans. *Nature Neuroscience*,

*15*(12), 1729–1735. https://doi.org/10.1038/nn.3248

O’Connell, R. G., Shadlen, M. N., Wong-Lin, K., and Kelly, S. P. (2018). Bridging Neural and Computational Viewpoints on Perceptual Decision-Making. *Trends in Neurosciences*,

*41*(11), 838–852. https://doi.org/10.1016/j.tins.2018.06.005

Pan, J., Klímová, M., McGuire, J. T., and Ling, S. (2022). Arousal-based pupil modulation is dictated by luminance. *Scientific Reports*, *12*(1), 1390. https://doi.org/10.1038/s41598022-05280-1

Patel, D., Fleming, S. M., and Kilner, J. M. (2012). Inferring subjective states through the observation of actions. *Proceedings of the Royal Society B: Biological Sciences*,

*279*(1748), 4853–4860. https://doi.org/10.1098/rspb.2012.1847

Szucs, D., and Ioannidis, J. P. A. (2017). Empirical assessment of published effect sizes and power in the recent cognitive neuroscience and psychology literature. *PLOS Biology*,

*15*(3), e2000797-. https://doi.org/10.1371/journal.pbio.2000797

Urai, A. E., Braun, A., and Donner, T. H. (2017). Pupil-linked arousal is driven by decision uncertainty and alters serial choice bias. *Nature Communications*, *8*. https://doi.org/10.1038/ncomms14637

Vafaei Shooshtari, S., Esmaily Sadrabadi, J., Azizi, Z., and Ebrahimpour, R. (2019). Confidence Representation of Perceptual Decision by EEG and Eye Data in a Random Dot Motion Task. *Neuroscience*, *406*(March), 510–527. https://doi.org/10.1016/j.neuroscience.2019.03.031

Wang, X. J. (2002). Probabilistic decision making by slow reverrberation in cortical circuits. *Neuron*, *36*, 955–968.

Wei, Z., and Wang, X.-J. (2015). Confidence estimation as a stochastic process in a neurodynamical system of decision making. *Journal of Neurophysiology*, *114*(1), 99– 113. https://doi.org/10.1152/jn.00793.2014

Wong, K.-F. (2006). A Recurrent Network Mechanism of Time Integration in Perceptual Decisions. *Journal of Neuroscience*, *26*(4), 1314–1328. https://doi.org/10.1523/JNEUROSCI.3733-05.2006

Yu, S., Pleskac, T. J., and Zeigenfuse, M. D. (2015). Dynamics of postdecisional processing of confidence. *Journal of Experimental Psychology: General*, *144*(2), 489–510. https://doi.org/10.1037/xge0000062